# Nano Carrier Drug Delivery Systems for the Treatment of Neuropsychiatric Disorders: Advantages and Limitations

**DOI:** 10.3390/molecules25225294

**Published:** 2020-11-13

**Authors:** Yana Zorkina, Olga Abramova, Valeriya Ushakova, Anna Morozova, Eugene Zubkov, Marat Valikhov, Pavel Melnikov, Alexander Majouga, Vladimir Chekhonin

**Affiliations:** 1Department Basic and Applied Neurobiology, V.P. Serbsky Federal Medical Research Centre of Psychiatry and Narcology, 119034 Moscow, Russia; abramova1128@gmail.com (O.A.); ushakovavm@yandex.ru (V.U.); hakurate77@gmail.com (A.M.); zubkov@ngs.ru (E.Z.); marat.valikhov@gmail.com (M.V.); proximopm@gmail.com (P.M.); chekhoninnew@yandex.ru (V.C.); 2Healthcare Department, Mental-Health Clinic No. 1 Named after N.A. Alexeev of Moscow, 117152 Moscow, Russia; 3Department of Biology, Lomonosov Moscow State University, 119992 Moscow, Russia; 4D. Mendeleev University of Chemical Technology of Russia, 125047 Moscow, Russia; rector@muctr.ru; 5Department of Medical Nanobiotechnology, Pirogov Russian National Research Medical University, 117997 Moscow, Russia

**Keywords:** nanocarriers, schizophrenia, bipolar disorder, depression, anxiety, Alzheimer’s disease, BBB

## Abstract

Neuropsychiatric diseases are one of the main causes of disability, affecting millions of people. Various drugs are used for its treatment, although no effective therapy has been found yet. The blood brain barrier (BBB) significantly complicates drugs delivery to the target cells in the brain tissues. One of the problem-solving methods is the usage of nanocontainer systems. In this review we summarized the data about nanoparticles drug delivery systems and their application for the treatment of neuropsychiatric disorders. Firstly, we described and characterized types of nanocarriers: inorganic nanoparticles, polymeric and lipid nanocarriers, their advantages and disadvantages. We discussed ways to interact with nerve tissue and methods of BBB penetration. We provided a summary of nanotechnology-based pharmacotherapy of schizophrenia, bipolar disorder, depression, anxiety disorder and Alzheimer’s disease, where development of nanocontainer drugs derives the most active. We described various experimental drugs for the treatment of Alzheimer’s disease that include vector nanocontainers targeted on β-amyloid or tau-protein. Integrally, nanoparticles can substantially improve the drug delivery as its implication can increase BBB permeability, the pharmacodynamics and bioavailability of applied drugs. Thus, nanotechnology is anticipated to overcome the limitations of existing pharmacotherapy of psychiatric disorders and to effectively combine various treatment modalities in that direction.

## 1. Introduction

According to statistics, one in four people in the world afflicts with various mental or neurological illnesses at some point of their lives. Around 450 million people currently suffer from such conditions, placing mental disorders among the leading causes of ill and disability worldwide [1].

There is a great clinical need for development a new-generation of psychoactive drugs, because the global burden of mental illness continues to increase. Despite the wide range of existing drugs with psychotropic and neurotropic activity, there has been no significant success in treating mental illness. This may be because it takes time to achieve an effective concentration of drugs, with which side effects are associated. This, in turn, forces clinicians to change their treatment methods frequently.

Preparations in their usual form (pills or injections) have many obstacles in their way to achieve the desired effect. These include poor bioavailability, low absorption (food effect), first-pass metabolism, drug toxicity and dose-dependent side effects.

The concept of nanotherapy includes systems with sizes of 10–100 nm and unique physiological and chemical properties. Nanopreparations are distinguished by their plasticity and modified surface properties, flexibility and controlled release of the active drug substance. Nanotherapeutics provide targeted delivery of central nervous system (CNS) active drugs to the brain [2], which is essential for drugs aimed at treating mental illness. An example of such preparations is biodegradable, biocompatible polymers, which are non-immunogenic and non-toxic to cells. Their increased solubility and targeted delivery result in an increased concentration of drugs in cells and tissues, thereby increasing the bioavailability of drug molecules. Therefore, nanoparticles (NPs) application demonstrates a variety of advantages in comparison with other methods of drug administration as intranasal, transdermal, oral and intravenous administration (Table 1).

Currently, various types of nanotherapy for mental disorders, including polymeric nanoparticles, solid lipid nanoparticles (SLN), nanostructured lipid carriers (NLC), nanoemulsions, nanogels, carbon nanotubes and liquid crystalline nanoparticles with neuroprotective properties [3] are being investigated.

In our review we will present current data on promising areas of nanotherapy in the field of mental and neurodegenerative illnesses, such as schizophrenia, bipolar disorder, depression, anxiety and Alzheimer’s disease.

## 2. Drug Transport through the Blood Brain Barrier

Despite significant progress in understanding the molecular and cellular mechanisms of brain disorders and the development of strategies for their treatment, efficient drug delivery to the central nervous system is an important problem today [4]. One of the most significant barriers in the central nervous system that interferes with drug delivery compounds is the blood brain barrier (BBB). Its functions include not only transporting nutrients and oxygen to the brain from the bloodstream, but also protection of the CNS from neurotoxic substances [5]. The BBB consists of several types of cells such as: endothelial cells, astrocytes, perivascular macrophages and pericytes [6]. The impermeability of the BBB for many molecules is due to the tight contacts between endothelial cells, which in turn form the proteins occludins, claudins and adhesion molecules [7]. Together with tight junctions of endothelial cells, a specialized extracellular matrix and a basement membrane from collagen IV type, laminin, fibronectin, tenascin and proteoglycans provide controlled BBB permeability [8]. The most studied pathways of substances through the BBB are the so-called passive or transcellular diffusion, facilitated diffusion and active transport. Transcellular transport of substances occurs through active and passive diffusion. In the case of active transport in the transfer of molecules, there are three types of transport systems in the BBB: receptor-mediated transport, which carries out the transfer of macromolecules; carrier proteins that transport sugars, amino acids, organic anions and cations, neurotransmitters and metabolites; active transport by proteins of the adenosine triphosphate binding cassette family [9]. Proteins of the adenosine triphosphate binding cassette family, such as P-glycoprotein, make the BBB impermeable to drugs [10].

One of the key problems today is the development of drugs for the treatment of CNS diseases, which will pass through the BBB. For effective therapeutic response the necessary concentration of the drug should accumulate in the right body region and remain maintained enough time to achieve that response. In other organs and tissues, the drug concentration has to stay minimal for side effects reduction. In that case, the presence of BBB significantly complicates the delivery of drugs into brain tissue.

To solve that problem, many methods have been developed to date, such as: violation of the integrity of tight contacts using hypertonic solutions (arabinose and mannitol) [11]; vasoactive drugs (leukotriene C4, IL-2, TNF-α, IFN-γ and bradykinin) [12] and directed ultrasound [13].

Other methods of overcoming the BBB are direct injection of drugs into the brain tissue. In these techniques, it is possible to distinguish intrathecal or intraventricular administration of drugs for direct delivery into the cerebrospinal fluid, intranasal administration and administration of biodegradable substances [14].

High concentrations of drugs in the brain can be achieved by intrathecal administration. Compared to systemic administration, this method allows the use of low doses of drugs and does not require changes in the integrity of BBB. However, this is an invasive strategy and can lead to increased intracranial pressure and neurotoxicity [15].

Intranasal delivery is another method of direct transport of a therapeutic agent for treating various diseases and CNS disorders. This method is based on the connection between the olfactory nasal lining mucosa and the circulation of cerebrospinal fluid next to the olfactory bulbs [16]. Intranasal delivery of pharmaceuticals is a non-invasive, safe and promising method of treatment.

One of the promising methods to overcome BBB is the loading of drugs into nanocontainers that are characterized by relatively small size and ability to cross the BBB. Using this method of drug delivery significantly increases substances bioavailability and reduces the side effects.

The mechanisms of NPs transportation through BBB can be classified into tree main types [17]:Transient opening of BBB is induced by a stimulus derived from bioactive components on the surface of nanoparticles, or by a stimulus derived from the “nanoeffects” or “nanotoxicity” of nanoparticles. This opening promotes the diffusion of drug conjugates into brain tissue;The adsorption of nanocarrier conjugates on the surface of capillary endothelium cells facilitates the release of the drug from carriers on the cell surface. This process increases the concentration gradient of the drug and promotes the diffusion of substances into the brain;Transcytosis, endocytosis and exocytosis of nanocarriers by capillary endothelial cells of the brain provides direct penetration of the substances into brain tissue.

Figure 1 describes the main pathways of nanoparticles entering the brain: mechanisms for BBB penetration and transport across the cell membrane.

Basically, there are three main mechanisms of NPs cellular uptake [18]. The first one is pinocytosis, which is divided into macro- and micropinocytosis, depending on the nanocarrier size and the type of targeted cell. Another method of nanotherapeutics uptake is clathrin-mediated endocytosis or receptor-mediated endocytosis [18]. This process is based on the formation of clathrin-coated endocytic vesicles. The third way to load NPs into cells is caveolae-dependent endocytosis with calveolar vesicles generation. This type of endocytosis is commonly found in endothelial and muscle cells [18].

In the brain different types of neural cells, including neurons, astrocytes, oligodendrocytes and microglia can represent a target for drug delivery. Targeting ability of nanotherapeutic agents depends on NPs functionality and sometimes requires the presence of specific molecules. Neuron-specific delivery is rather difficult because neurons have a non-phagocytic nature in comparison to glial cells. In some studies, neuron-specific targeting was achieved by the properties of connecting ability of loaded substances. To enhance the neuronal targeting, active strategies such as utilizing ligands with dopamine [19], and rabies virus glycoprotein peptide (RVG and RDP peptide) [20] have being used. Receptor-mediated endocytosis strategies are commonly used to induce neuron uptake of NPs. For, example, nanocarriers coated with apolipoprotein E can be uptaken by neurons through the agency of the lipoprotein-binding receptors on neurons surface [21].

Astrocytes are also very promising targets for nanomedicine. Commonly NPs can be transported into astrocytes via phagocytosis. Their targeting can affect the neurotransmitters release (e.g., glutamate, γ-aminobutyric acid, glycine and histamine), which is crucial for the maintenance of neuronal excitability [22].

Oligodendrocytes are specialized neural cells, where their role is to myelinate the axons in CNS. Properties of their surface can simplify drug delivery via nanopreparations. In particular, NG-2 chondroitin sulphate proteoglycans are expressed on oligodendrocyte precursor cells, which represent useful targets for specific NPs like poly(lactic-co-glycolic acid (PLGA). Such nanocarriers loaded with leukemia inhibitory factor can increase myelin repair [23].

Targeting of microglia or macrophages is also a promising direction for nanomedicine. It is known, that clathrin-mediated endocytosis serves as the major pathway for microglia to uptake NPs. NPs with a larger surface area demonstrate higher uptake efficiency, because clathrin-mediated endocytosis is carried out by the binding of clathrin receptor with the surface of nanocarriers. This property was manifested by the experiments demonstrated that gold NPs with “urchin-mimicking” geometry were absorbed more effectively by microglial cells compared to spherical and rod-like gold NPs [24]. In addition to clathrin-mediated endocytosis, mannose receptors and macrophage scavenger receptors have also been implicated in playing a role in nanocarriers uptake in non-activated microglia [25,26]. There are two main ways to affect microglia or macrophages with nanotherapy. The first one allows controlling microglia/macrophages population by inhibition of infiltrating monocytes/macrophages. For these reason cytotoxic drugs, such as clodronate, glucocorticoids, doxorubicin and methotrexate are usually applied. For example, it was demonstrated that locally delivered methylprednisolone encapsulated into PLGA nanoparticles dramatically reduced the number of macrophages/reactive microglia [27]. The second way to modulate glial cells functions is the use of control release of cytokines via NPs.

Therefore, various research groups are developing nanocontainer systems for drug delivery. There are several types of using nanoparticles with different advantages and disadvantages. In the following we will consider the types of nanoparticles and their way of penetration through BBB, which is necessary for the treatment of neuropsychiatric disorders.

## 3. Categories of Nanocarriers

Currently the spectrum of nanoparticles used for targeted drug delivery is quite wide. Such nanocontainers can be divided into three main groups: polymeric nanoparticles (NPs), lipid-based NPs and inorganic NPs (Table 2).

### 3.1. Polymeric Nanoparticles

This class includes polymer-based nanoparticles and micelles and dendrimers (Table 2).

Nanoparticles based on polylactide acid (PLA) and poly (lactic-co-glycolic acid) (PLGA) developed over several decades have proved to be a promising alternative system for delivering drugs to the brain, primarily due to their high biocompatibility and low toxicity. They have uncomplicated synthesis methods (emulsification, evaporation, solvent replacement method (deposition), solvent diffusion method, etc.) and are metabolized into lactic and glycolic acids using a simple mechanism. So, they are biodegradable, with optimal sizes ranging from 100 to 200 nm. PLGA/PLA nanoparticles are U.S. Food and Drug Administration (FDA) certified and can be used in implants, biodegradable scaffolds and human medicines.

The therapeutic molecules encapsulated in polymer nanoparticles can form the following thermodynamically stable structures, depending on the loading method. Either they are nanospheres where the drug is loaded across the entire polymer matrix or only on its surface, or they are nanocapsules where the active substance is surrounded by a polymer shell.

The surface of such particles is easily modifiable, both with ligands for target delivery and molecules that promote adsorption on target cells. Thus, using the example of the anticonvulsant Clonazepam contained in the PLGA matrix, it was possible to achieve effective delivery of the drug through BBB by conjugating the poloxamer-188 to the surface of the NP [35]. Conclusion in PLGA nanocarriers curcumin solved the problem of its insolubility in water, and conjugation with Tet-1 made it possible to increase nanoparticles capture in the treatment of Alzheimer’s disease (AD) [36].

Polymer nanoparticles penetrate the BBB through endocytosis. They can also avoid the phagocytosis of the reticular endothelial system, thereby increasing the concentration of drugs in the brain [37]. In vitro studies have shown that the use of polymer nanoparticles increases the delivery of drugs to the brain. For example, improved delivery of curcumin (in the treatment of AD) is more effective in reducing oxidative stress, inflammation and plaque load.

#### 3.1.1. Dendrimers

Dendrimers are monodisperse symmetrical macromolecules consisting of a series of branched blocks around the inner nucleus. The spherical structure of dendrimers consists of the core, layers of branched, repeating blocks coming out of the core and functional end groups on the outer layer [17].

In the treatment of brain diseases, polyamidoamine dendrimers are most commonly used among dendrimers. For example, there is a study on the use of encapsulated carbamazepine (an antiepileptic drug) for the treatment of AD [38].

#### 3.1.2. Micelles

Micelles consist of amphiphilic block copolymers. They aggregate in aqueous solutions to form stable spheroidal nanostructures that have a hydrophobic core and hydrophilic surface.

The micelles are promising agents for delivering drugs to the brain. This is made possible by the dissolution of poorly soluble substances in the hydrophobic nucleus area and the ability to conjugate with certain target ligands [17].

### 3.2. Lipid-Based NPs

This type of nanoparticles includes liposomes, solid lipid nanoparticles, nanoemulsions and exosomes (Table 2).

#### 3.2.1. Liposomes

Liposomes are the family of nanostructures, the main characteristic of which is a spherical shape and lipid bilayer. Optimal sizes range from 100 to 200 nm. The first FDA approved nanodrug in 1995 was Doxil represented a preparation based on liposomes [39]. Over the next 2 decades all types of liposome forms, methods of construction and modifications have been developed [40]. For example, liposomal nanoparticles produced by microfluidics were constructed. These nanocarriers allow increasing the percentage of drug loading and obtaining a balanced size of nanocontainer [41]. Generally, liposomes are characterized by the high biocompatibility, biodegradability, improved bioavailability and stability of therapeutic agents [42]. Moreover, they are able to modify the surfaces for active targeting [43]. Nanoparticles structure allows the incorporation of auxiliary lipids such as dioleylphosphatidylethanolamine (DOPE) or cholesterol. DOPE is able to conjugate with other lipids when exposed to low pH, which promotes increased efficiency of loading into cytosol and favor endosomes construction [44]. Cholesterol provides structural stability. The linkers are sensitive to different biological stimuli and contribute to the release of the drug under certain conditions. For example, modifications of gold nanoshells liposomes mediating by chitosan can provide liver protection [45]. Opsonization and capture by macrophages of the organs of the mononuclear phagocytic system obstructs the delivery of drugs via unmodified liposomes. Modification with polyethylene glycol (PEG) increases nanoparticles circulation time and reduces cell capture [17].

The using of liposomal constructions for BBB crossing has been investigated for decades [43]. Liposomes are modified for targeting transcytosis through CNS endothelium via the liposomal surface functionalization with biologically active ligands, such as peptides, antibodies or small molecules. G-Technology^®^ (FDA approved technology) is another method for transformation of the liposomal surface. It represents a strategy for using glutathione-PEGylated liposomes, which are able to bind with glutathione transporter [46].

Thus, liposomes are promising candidates for BBB penetration and targeting delivery of therapeutic agents.

#### 3.2.2. Solid Lipid Nanoparticles

Unlike liposomes, this type of nanoparticles does not possess a lipid layer, but is characterized by the presence of a solid lipid core matrix that can encapsulate lipophilic molecules. Solid lipid nanoparticles (SLNs) are nanoscale suspensions of biologically compatible lipids such as triglycerides, fatty acids or waxes stabilized by bioactive substances possessing a hydrophile-lipophile balance [17].

Such nanoparticles are also subjected to surface modification, which significantly increases the efficiency of drug transporting through BBB [47]. As containers for drug delivery, they demonstrate a high level of biocompatibility, biodegradability and the ability to transform the surface with target bioactive molecules. However, their hydrophobicity promotes easier clearance by the reticuloendothelial system (RES), which alleviates nanoparticles effectiveness.

#### 3.2.3. Nanoemulsion

Nanoemulsions are heterogeneous dispersions of the composition “oil in water” (O/W) or “water in oil” (W/O), stabilized by surfactants. The diameter of the internal phase in such nanosystems is reduced to a nanometric scale. Due to their properties, nanoemulsions demonstrate the ability to carrier both hydrophobic (O/W) or hydrophilic (W/O) drugs. The surface of these nanoparticles can also be modified with bioactive ligands. Delivery of drugs via nanoemulsions is carried out through receptor-mediated endocytosis of cells.

Compared to other nanocarriers, the preponderance of nanoemulsions for BBB penetration is the ability to use safe oils. In addition, the application of this type of nanoparticles possesses a number of useful biological properties due to the presence of essential omega-3 and omega-6 fatty acids in the oils [17].

### 3.3. Inorganic NPs

Inorganic nanoparticles represent solid nanosize objects consisting of inorganic materials such as silicon dioxide, carbon nanotubes, metal or metal oxide. This type of nanoparticles is widely used for visualization research, including neuroimaging.

To facilitate penetration through BBB, they are covered with several polymers such as polysaccharides, polyacrylamide, poly(vinyl alcohol), poly(*N*-vinyl-2-pyrrolidone), PEG and PEG-containing copolymers. The most commonly used covering polymer is PEG. These surface modifications enhance nanoparticles stability, improve water solubility and allow modifying of the particle surface with vector molecules [48]. In addition, such PEGylation of nanocontainers averts the capture of particles by the RES and prevents non-specific interactions with opsonin proteins [49].

Besides PEG covering inorganic nanoparticles are also modified with lactoferrin (Lf). It is interesting that this conjunction demonstrated better efficiency during BBB penetration than PEG because of Lf-receptor-mediated transcytosis of cerebral endothelial cells [50].

In addition to chemical modification, the physical properties of inorganic nanoparticles can also be changed for an increase of penetration through BBB. For example, Fe_3_O_4_ nanoparticles demonstrate the ability to be magnetized in the presence of a magnetic field. It has been shown that superparamagnetic iron oxide nanoparticles (SPIONs) contained magnetite (Fe_3_O_4_) and maghemite (c-Fe_2_O_3_) can cross human brain microvascular endothelial cells via an external magnet pressure. Therefore, this mechanism demonstrates the ability of BBB penetration and targeted drug delivery to the brain to be carried out by the magnetic force-mediated dragging of SPIONs through the BBB. Such property of iron oxide nanoparticles as magnetism can be applied for diagnostics and simultaneous therapy of neurodegenerative diseases [51]. Along with iron nanocarriers gold nanoparticles can also penetrate through BBB and can be administered for therapy of neuropsychiatric diseases [52].

Nevertheless, despite the advantages of these types of nanoparticles, they exhibit potential toxicity due to their nondegradability. For this reason, their application requires further research.

## 4. Drug Delivery for Depression

Depression is a widespread disorder, affecting almost 350 million people worldwide [53]. Depressive symptoms influence quality of life of patients and aggravate the course of comorbid diseases. Depression pathogenesis affects various neurobiological processes including HPA axis activity, monoamine systems functioning, neurogenesis, neuroinflammation, etc. [54]. Traditional medications for the treatment of depression are antidepressants with different mechanisms of action affecting monoamine neurotransmission in the brain.

Antidepressants are divided into several classes according to its mechanisms [55].

Tricyclic antidepressants (TCAs) block the reuptake of monoamine neurotransmitters (primarily serotonin and norepinephrine) presynaptically.Monoamine oxidase inhibitors (MAOIs) amplify concentration of monoamines in the synaptic cleft by the blockage of monoamine oxidase enzymes.Selective serotonin reuptake inhibitors (SSRIs) selectively block the reuptake of serotonin. This group of drugs demonstrates decreased side-effects in comparison to other classes of antidepressants.Serotonin and norepinephrine reuptake inhibitors (SNRIs) produce dual inhibition of serotonin and norepinephrine reuptake.Norepinephrine-dopamine reuptake inhibitors block the action of norepinephrine and dopamine transporters thereby inducing reuptake suppression.Norepinephrine reuptake inhibitors block norepinephrine and epinephrine reuptake via norepinephrine transporter inhibition.Serotonin antagonists and reuptake inhibitors (SARIs) act by antagonizing serotonin receptors such as 5-HT2A and inhibiting the reuptake of serotonin, norepinephrine and/or dopamine. Most of them also act as α1-adrenergic receptor antagonists. The majority of the currently marketed SARIs belong to the phenylpiperazine class of substances.NMDA receptor antagonists deactivate NMDA glutamate receptors. This class of antidepressants characterizes as rapid-action drugs and includes such substances as ketamine and esketamine.

In spite of the variety of therapy methods, depression treatment is still a serious concern among psychiatrists. A major study by the National Institute of Mental Health found that less than 50% of people are completely recovered from symptoms of depression even after taking two different antidepressants. Moreover, many of those who respond to treatment soon become depressed again, even though they continue to take their medication [55].

Effectiveness of antidepressant therapy depends on the concentration of the drug in the brain. Conventional oral and parenteral therapies are limited because it requires the drug to cross the BBB.

Antidepressants in dosage forms that manageably release the drug directly into the brain reduce their systemic effects, side effects, drug interactions and ultimately bypass pharmacoresistant problems.

At present, quite a few research groups are busy studying the pharmacokinetics, pharmacodynamics and efficacy of antidepressant nanodrugs. Their research are at various stages: from nanoproducts design to preclinical and clinical stages of testing. So far, none of the antidepressants with targeted delivery to the brain have passed the FDA.

Literature analysis suggests that the intranasal administration of nanotherapy (where the drug is encapsulated in nanoparticles) is the most advantageous way to deliver a whole set of antidepressants of different classes to the brain [56,57]. Moreover, this is consistent with the assumption that a faster start of action accompanied by prolonged circulation of active molecules increases efficiency. Polymer nanoparticles are the most frequently investigated among the various nanosystems (Table 3).

### 4.1. TCAs

Doxepin is one of the tricyclic antidepressants used for depression treatment. During the last few years doxepin was also investigated as a nanoform. In particular, doxepin in the form of thermoreversible biogel or chitosan and glycerophosphate or polyethylenenglycol demonstrated lower local toxicity and significant effectiveness in the forced swim test [113].

### 4.2. MAOIs

Antidepressants from MAOIs class have been also investigated for targeting drug delivery. For example, a selective inhibitor of MAO type B—Selegiline was loaded into nanoparticles for depression treatment. It was demonstrated that thiolated chitosan nanoparticles developed by Singh et al. (2016) enhanced the nasal delivery of selegiline and produced antidepressant-like effect in rats [69].

### 4.3. SSRIs

SSRIs have also being studied as potential nanopreparations. Paroxetine as nanoemulsion form [114] and fluoxetine as ion sensitive in situ nasal gel [115], showed significant responses in locomotor activity level and immobility in injected rats compared to control groups.

Citalopram is also one of the SSRIs. It belongs to the few drugs that can be used safely in childhood psychiatric disorders treatment. Along with other SSRIs Citalopram was loaded in interpenetrating polyelectrolytes nanocomplexes (composed of chitosan: pectin in a 3:1 ratio) constructed by Kamel and colleagues [116]. During experiment researchers demonstrated extended drug release pattern, higher serotonin brain level up to 24 h, more rapid onset of action and more extended behavioral effects in rats with depressive-like behavior treated with these complexes compared to pure citalopram.

Sertraline is another SSRI drug characterized by poor water-solubility and low oral bioavailability. Sertraline-loaded solid lipid nanoparticles improved the drug releasing. In particular, the maximum plasma concentration after nanoparticles injection was 10-fold higher compared to pure drug. Moreover, Sertraline-loaded SLN were found to be stable at 4 °C for 6 months of the study period [117].

### 4.4. SNRIs

One of the promising antidepressants for targeting delivery is venflaksin hydrochloride (SNRI). There are several ways to deliver it into the brain, for example by Lutrol F127 [118], nanoparticle-based NLC gel [56], alginate–chitosan nanoparticles (AGNPs) [119]. It was shown that nanoparticles loaded with venlafaxine hydrochloride produced a more pronounced effect after intranasal administration in vivo in comparison to oral administration of equivalent dose [118,119]. Desvenlafaxine is another preparation that belongs to that class of drugs. It represents an active metabolite of venlafaxine also investigated in PLGA-chitosan nanoparticles [120].

Duloxetine (SNRI) is another antidepressant that demonstrated better response in NLC [47]. The scintigraphy images are consistent with the biodistribution and pharmacokinetic studies, having revealed a high uptake of duloxetine into the rat brain [47].

### 4.5. SARIs

Serotonin antagonists and reuptake inhibitors are drugs characterized by antidepressant, anxiolytics and hypnotics effects. This class of substances includes trazodone and nefazodone. These drugs were also loaded into nanoform. In particular, Casolaro et al. (2015) constructed hydrogels bearing l-phenylalanine (Phe-Nip3) or l-valine (Ava2) residues, with citalopram and trazodone. Hydrogel form showed improved release of these drugs compared to pure substances [121].

### 4.6. Other Drugs

Agomelatine is atypical antidepressant that both stimulates melatonin receptors (MT1 and MT2) and blocks 5-HT2C-receptors. This drug is characterized by low side-effects because it does not counteract with other serotonin receptors, does not affect reuptake of monoamines and does not interact with adrenergic, cholinergic, dopamine, histamine and GABA receptors. Nevertheless, agomelatine has a low biological half-life coupled with extensive hepatic first-pass metabolism [59]. In that context the loading of preparation into nanoform could improve its therapeutic effect. Thus, it was demonstrated that intranasal administration of Agomelatine-loaded poly-lactic-co-glycolic acid nanoparticles shown significant antidepressant-like effect in rats [58]. Another type of nanoparticles was developed by Shinde et al. (2020) and represented polymeric nanoparticles with Agomelatine for transdermal use. These nanocarriers demonstrated good permeability across rat skin [59].

Another type of irregular antidepressant is flavonoid Baicalein. Chen and colleagues constructed solid lipid NPs loaded with Baicalein [60]. These NPs were modified with *N*-Acetyl Pro-Gly-Pro (PGP) peptide, which is characterized by high specific binding affinity to neutrophils through the CXCR2 receptor. When associated with neutrophils, these particles can penetrate through the BBB and reach targets in the brain.

Along with research of classical antidepressants in nanoforms, it is quite interesting to study various peptides and neurotrophins as antidepressant treatment. In this regard, challenging data concern the proven neuroprotective and antidepressant properties of the brain-derived neurotrophic factor (BDNF). In vitro experiments showed that BDNF delivery using the adenoviral vector produce neuroprotective effects. AAV-Syn-BDNF-EGFP virus vector maintains brain cell viability and preserves the neural network functional structure in the late posthypoxic period [122].

It was found that intranasal administration of BDNF-HA2TAT/AAV to normal mice produce an antidepressant effect in the forced swimming test, but only with chronic treatment. No statistical difference was found in the forced swim test after acute intranasal injection of the AAV to the mice. After the BDNF-HA2TAT/AAV administration to the animals with the chronic mild stress model of depression the successful therapeutic effect was demonstrated in the tail suspension test and forced swim test, but not in open field test [123]. Moreover, the female mice were more sensitive to the drug administration. One of the possible explanations is that the prevalence of depression in females is much higher than that in males [124].

Therefore, different nanocontainer forms of drugs have been developed for the treatment of depression. Such systems contain not only the most popular antidepressants, such as SNRI and SSRI, but also various original forms including vector systems for target delivery in the brain. Application of nanocarriers could help to reduce side effects and to improve drugs bioavailability.

## 5. Drug Delivery for Anxiety Disorders

Anxiety disorders are the largest group of mental disorders in most Western societies. Essential features of anxiety disorders are excessive and persistent fear, avoidance of perceived threats and panic attacks in certain instances. Although the pathophysiology of anxiety disorders is not fully understood, targeting systems for their treatment are of interest. Drugs for treating anxiety disorders include medications that affect serotonergic, adrenergic, glutamatergic, endocannabinoid systems and various neuropeptides [125]. First of all, tranquilizers with anxiolytic, myorelaxing, sedative and anticonvulsant activity are used for the treatment of anxiety disorders. Many anxiolytic drugs are derived from benzodiazepine and affect benzodiazepine receptors (second generation anxiolytics). However, there is also a spectrum of tranquilizers aimed at other therapeutic targets. Examples include such drugs as buspirone, meprobamate, benactin, etc. Significant progress has been made in the development of formulations and psychosocial treatment strategies for anxiety disorders. However, symptoms persist in many patients despite improvements in therapy. Randomized controlled trials of anxiety disorders pharmacotherapy report a response rate of 40–70% and a remission rate of 20–47%. Anxiety disorders are considered as resistant to treatment when there are residual symptoms or when symptoms do not improve at all after some form of therapeutic intervention. The optimal use of available medication and the development of new pharmacological strategies and agents promise to improve this situation [126].

Polymer nanocarriers are used to develop delivery forms for therapeutic drugs against anxiety disorders. Nanoparticles created from biodegradable polymers such as PLGA are widely studied as drug delivery systems because of their two important properties, such as biocompatibility and controlled drug release. For example, replicable spherical diazepam PLGA nanoparticles have been developed and characterized by Bohrey and colleagues [81]. Chitosan is another natural carbohydrate polymer with ideal for nanoparticles polymer carrier properties such as biocompatibility, biodegradability, non-toxicity and low cost. It exhibits bioadhesive properties and the ability to significantly increase permeability for hydrophilic compounds. Chitosan with thiol groups provides much higher adhesion properties.

Buspirone is a drug belonging to the group of tranquilizers using for anxiety disorders treatment. It is a complete agonist of 5-HT1A presynaptic serotonin receptors, which leads to a decrease of serotonin release and the resulting reduction of anxiety symptoms. Buspirone also acts as a partial agonist of the post-synaptic serotonin receptors, which contribute to the release of serotonin in synapses, resulting in increased serotonin concentration and reduced depression. Additionally, buspirone inhibits the dopamine receptors D2 [127]. Bari et al. (2015) loaded buspirone hydrochloride into polymeric thiolated chitosan nanoparticles for intranasal administration. Ex vivo study carried out via the porcine mucin binding efficiency test demonstrated excellent bioadhesion. Moreover, the drug concentration in the brain after the intranasal injection was significantly higher than after pure Buspirone hydrochloride administration [79]. Along with chitosan nanoparticles construction two types of nanovesicular in situ gels (based on carbopol 974P and poloxamer 407) were developed for delivery of buspirone hydrochloride. The carbopol composition showed higher permeability compared to the poloxamer. It was demonstrated that buspirone hydrochloride has low bioavailability of about 4% after oral administration due to first pass metabolism. Nevertheless, the bioavailability of buspirone hydrochloride increased by 3.26 times when used in the composition of the nanovesicular gel of carbopol compared to the nasal solution of a pure drug [80].

Gallic acid has significant anxiolytic activity, which may be due to its antioxidant properties and reduced nitrite levels in plasma. Chitosan nanoparticles coated with Tween 80 ligand were used to deliver gallic acid to the brain. Behavioral studies have shown that GA nanoparticles significantly improve anxiolytic activity in mice. Nitrite levels decrease in plasma was most pronounced in the group that received the nanoparticle with loaded with drugs. This may have been due to the reduction in plasma nitrite levels by gallic acid. This effect was improved by the increase of GA brain absorption through the nanoparticles application [82].

Inorganic nanoparticles are also used to create delivery systems for anxiety disorders. Many peptides and their analogues cannot effectively cross the BBB. For that reason, Vinzant and colleagues conjugated peptide antisauvagine-30 (ASV-30) with iron oxide nanoparticles [78]. It was shown that ASV-30 peptide affects corticotrophin releasing factor type 2 receptors when directly injected into the brain and thus reduces anxious behavior in animals. Imaging studies demonstrated that iron oxide + ASV-30 particles were present in the brain and were associated with neurons, including those that express CRF2 receptors after intraperitoneal injection. Brain distribution studies were affirmed nanoparticles anxiolytic activity in rats [78].

Currently, the range of application of anxiolytic drugs and their assortment are quite wide. These drugs include preparations of benzodiazepine-line and non-benzodiazepine anxiolytics. Nevertheless, not many nanocontainer forms have been developed for the standard medicines using for anxiety disorders treatment. The loading of other types of drugs in nanocarriers besides the usual forms of anxiolytics could help to improve the treatment of anxiety disorders and to prevent aggravation into depression or severe generalized form.

## 6. Drug Delivery for Schizophrenia and Bipolar Disorders

Schizophrenia is a complex endogenous mental disorder characterized by cognitive and emotional symptoms that significantly reduce the quality of life of patients [128]. The main drugs used to treat the pathology are antipsychotic drugs that reduce the increased dopaminergic transmission in the brain of patients. Atypical antipsychotics of a new generation, characterized by a narrower range of side effects compared to the first generation, are also used to treat bipolar disorder, along with lithium and antiepileptic drugs [129].

The first drug for the treatment of schizophrenia, chlorpromazine, was first used in 1952, but already in the 60s researchers tried to develop long-acting substances consisting of an antipsychotic agent and fat-soluble solution [130].

Despite the fact that new antipsychotic drugs have appeared on the market, providing greater efficiency with significantly reduced side effects, delivery systems of antipsychotic drugs are still in the stage of conventional drug forms, such as tablets, capsules and solutions, and need to be dosed 2–4 times a day. There is no doubt that new drug delivery systems, such as stable and controlled release systems, will be useful for antipsychotics. They should reduce the frequency of dosing, increase bioavailability of the drug and improve adherence of the therapy course [131].

The drug delivery system produces less pronounced side effects than the original drugs, because they provide lower inpatient concentrations of therapeutic drugs and more stable release characteristics than oral drugs. Thus, such combined drugs increase exposure to oral medications, further reducing morbidity and mortality [132]. As nanocarriers bypass the gastrointestinal tract, such a type of therapy reduces the amount of medication needed and can minimize some peripheral side effects, including hepatotoxicity and hyperprolactinemia [130].

Nanocontainer systems based on PLA were developed in 1989 for chlorpromazine—the first antipsychotic agent [131].

Since then atypical neuroleptics have appeared and become widespread in medical practice. A lot of publications about nanocontainers involve drugs with olanzapine—a second-generation or atypical antipsychotic, which selectively binds to central dopamine D2 and serotonin (5-HT(2c)) receptors. It has poor bioavailability due to hepatic first-pass metabolism and low permeability into the brain due to efflux by P-glycoproteins [133]. Due to its low oral bioavailability, researchers have created various container preparations [134]. Joseph et al. 2017 conducted research on glyceryl monostearate nanoparticles with olanzapine [133]. A longer release of the drug within 48 hours was shown, compared with 8 hours of the antipsychotic action of pure olanzapine solution. Pervaiz and colleagues described the release of olanzapine loaded in PNA (poly(*N*-isopropylacrylamide-co-acrylic acid)) microgels for 72 h [135]. In vitro bioaccessibility research of Jawahar and colleagues indicated 5½-fold increase in oral bioavailability for nano structured lipid carriers with olanzapine [134]. Cyclodextrins containers were also used for this neuroleptic [136]. An in vitro study on kinetics of clozapine solid lipid nanoparticles showed an increase in lymphatic bioavailability of the drug and its steady release up to 48 h [137]. The study done by Shafaat and coworkers demonstrated that clozapine nanoemulsions resulted in 4.4-fold increase in bioavailability and better drug loading capacity of hydrophobic drug molecules compared with marketed drug formulation [138]. There are also studies on the intranasal introduction of nanoparticles with olanzapine [94]. Nanoparticles containing olanzapine (PLGA) have been shown to be more effectively delivered to the brain by intranasal administration of poly(lactic-co-glycolic acid) nanoparticles [97]. Olanzapine-SLNs showed a 23-fold increase in the relative bioavailability of olanzapine in the brain after nasal administration [92]. The innovative therapeutic approach represents a new form of pharmaceuticals—transdermal drug delivery systems—“patches” containing lipid nanocontainers with this neuroleptic [139]. In addition to improving bioavailability, a study [140] with biodegradable polymeric lipid-core nanocapsules and a decrease in extrapyramidal side effects were shown for polymeric nanoparticles polycaprolactone [141]. Animal model studies have demonstrated improved therapeutic efficacy compared to pure olanzapine [141].

Lurasidone (another atypical antipsychotic) was loaded into SLNs. Its improved therapeutic effect in MK-801 induced schizophrenia model in rats after oral administration was shown [89].

Aripiprazole, a second-generation (or atypical antipsychotic), is poorly soluble and undergoes extensive hepatic metabolism and P-glycoprotein efflux, which lead to reduced in vivo efficacy and increased dose-related side effects. Solid lipid nanoparticles with aripiprazole enhance bioaccessibility of this neuroleptic [83]. In another study, intranasal administration of poly-caprolactone nanoparticles showed a 2-fold increase in aripiprazole distribution in rat brain when compared to aripiprazole nanoparticles administered intravenously [84]. An interesting study was devoted to buccoadhesive chitosan films with aripiprazole. Ex vivo nanocrystal loaded buccal films FAPZ 14 have the potential to provide a faster availability of aripiprazole. The aim of the present study was to design a mucoadhesive dosage form for buccal delivery of aripiprazole, which could provide a rapid drug delivery to the systemic circulation [142].

The same mucoadhesive drug delivery systems were created for riperidone, another atypical neuroleptic used to treat schizophrenia and bipolar disorder. A mucoadhesive buccal tablet formulation of risperidone allowed 90% release of the drug over 8 h, thereby providing direct systemic delivery of risperidone [143]. Risperidone-loaded solid lipid nanoparticles drug release and transport studies illustrated the advantages of oral delivery over poorly water-soluble drugs such as risperidone [144,145]. Pharmacokinetic studies of NLCs of iloperidone 2-fold increase in the % drug targeting efficiency relative to iloperidone pure drug suspension [146]. The pharmacokinetic study revealed almost 8.30-fold increase in oral bioavailability of iloperidone NLCs as compared to the plain drug solution. The nanoemulsions of riperidone also showed improved bioavailability and brain uptake [147]. Intranasal administration of risperidone-loaded mucoadhesive nanoemulsion demonstrated higher efficacy, enhanced brain targeting and higher drug transport to the brain in Swiss albino rats, when compared with intranasal risperidone solution and risperidone nanoemulsion intravenously [105]. In another study, riperidone was enclosed in biodegradable proteinoids. These systems were synthesized by thermal step-growth polymerization from the amino acids l-glutamic acid, l-phenylalanine and l-histidine and poly (l-lactic acid). Behavioral studies on mice found enhanced antipsychotic activity compared to free risperidone.

Quetiapine fumarate, a second generation atypical antipsychotic drug has a plasma half-life of 6h, hence requires frequent administration to maintain effective therapeutic concentration. The increase in bioavailability of mucoadhesive microemulsion Quetiapine during intranasal injection is shown [111].

Azenapine appeared on market in 2009. It was a new atypical antipsychotic for the treatment of schizophrenia and bipolar disorder. However, it had the same disadvantage—low bioavailability when used orally (less than 2%). Most of this drug is metabolized in the liver. The antipsychotic activity of azenapine is thought to be due to antagonistic activity on D2 dopamine and 5-HT2A serotonin receptors. To improve its bioavailability some of the researchers developed sublingual film, intranasal and injectable formulations of azenapine [86]. Shreya et al. (2016) used transfersomes, deformable elastic liposomes loaded with azenapine and transdermal plasters, as a drug formulation. Transfersomes as a drug carrier showed increasing the transdermal permeation and bioavailability of azenapine during in vitro and in vivo experiments.

Lithium medications—psychotropic drugs from the group of normotimists—are historically the first drugs of this group, discovered in 1949. However, they remain essential in the treatment of affective disorders, primarily manic and hypomaniacal phases of bipolar disorder [148]. They are also used to prevent its exacerbation and to treat severe and resistant depression. Their important property is to get rid of suicidal thoughts, so they can prevent suicide.

Lithium ions have a variety of effects on the nervous system, particularly as an antagonist of sodium ions in nerve and muscle cells. In this way, they weaken the nerve impulse (this also explains one of the frequent side effects of lithium preparations: muscle weakness). Lithium also affects the metabolism and transport of monoamines (noradrenaline and serotonin) and increases the sensitivity of some areas of the brain to dopamine. According to some data, the main role in the mechanism of action of lithium is its ability to block the activity of enzymes involved in the synthesis of inositol, which plays a role in regulating the sensitivity of neurons.

The treatment with lithium bipolar disorders is based on its property to selectively inhibit kinase 3 glycogen synthase activity [149]. In addition, it was found that in mania, there is an irregular increase in the activity of proteinkinase C, and a recent study has shown that lithium inhibits its activity. Thus, lithium, like other PkC inhibitors, exhibits antimaniac properties.

Formulations with lithium are the standard drugs for the treatment of bipolar disorders [148]. In one work, the researchers concluded lithium carbonate in chitosan nanoparticles to improve bioavailability. It is postulated that the encapsulation of lithium carbonate within chitosan nanoparticles enables the drug to overcome the intracellular degradative blocks such as endosomes and lysosomes. The ion chelation route was used for encapsulating lithium carbonate [88].

Neuroleptics therapy frequently used for schizophrenia and bipolar disorder treatment requires constant use of drugs and thereby induces various side effects. The loading of drugs into nanocontainers for the controlled release is modern and relevant approach. In addition to targeted delivery and increased bioavailability, it reduces the doses and side effects of antipsychotics, often accompanied its application.

## 7. Drug Delivery for Alzheimer’s Disease

Currently, the most intensive development of drugs based on nanocontainer systems is being carried out to treat AD. The progress of modern medicine has significantly increased average life expectancy. At the same time, the risks of neurodegenerative diseases rise with age. For example, the percent of people suffering from the most frequent neurodegenerative pathology, AD, in the ages from 65 to 74 is 3% and in the ages from 75 to 84 is already 17%. AD is a progressive disorder characterized by memory, cognition and behavioral impairment (dementia), which eventually leads to mood fluctuation and fatal delirium [150].

The action of the existing drug groups is directed on the pathogenesis blocks presumed as the hypotheses of AD pathogenesis.

Hypotheses of AD development can be divided into several main causes [32,151].

Accumulation of β-amyloid (Aβ(1 → 40) and Aβ(1 → 42)), which is resulted from pathologic proteolysis of amyloid precursor protein (APP). The aggregates of this protein are toxic and induce neurodegeneration, cytotoxicity and inevitably leading to dementia manifestations.Tau accumulation. It is known that a normal mature neuron produces three microtubule-associated protein (MAP) taus; MAP1A, MAP1B and MAP2. These proteins are responsible for the promotion of the assembly and stability of microtubules. The biological activity of tau is regulated by its phosphorylation level. For example, in the brain of AD patients, tau hyperphosphorylates abnormally, which impairs its binding to microtubules; leading to the accumulation of neurofibrillary tangles and dementia development. Several anti-tau therapies including lithium, valproate and nicotinamide have been already investigated for the prevention of Tau protein hyperphosphorylation.Progressive decline of acetylcholine concentration. This is the earliest hypothesis of AD progression. It was demonstrated that AD is accompanied by substantial presynaptic cholinergic deficit, which expresses as a reduced choline uptake and acetylcholine release in the brain. That leads to memory and cognition impairment. Drugs that eliminate these symptoms include acetylcholinesterase inhibitors (AChEs) such as tacrine, donepezil, rivastigmine and galantamine.NMDA excitotoxicity. Some experimental and neuropathological evidence suggests that glutamatergic system is also involved in the neurodegenerative progression. NMDA receptor is essential for the control of synaptic plasticity and memory function. It is activated by glutamate and glycine, allowing Ca2+ and Na+ influx. It was demonstrated that the hyperexcitability of NMDA receptors induces Ca2+ overload, eventually leading to apoptosis and producing cognitive impairments. The drug that affects this pathogenesis block is memantine (NMDA inhibitor).Abnormality of mitochondria energy metabolism. It is suggested that genetic mutations alter the regulation of the electron transport chain complex enzymes, which are capable of generating ROS. That leads to cell apoptosis and neurodegeneration.

Figure 2 describes the main pathological pathway of Alzheimer’s disease and therapy methods, targeted to these mechanisms.

Currently, nanoparticles with various therapeutic mechanism associated with AD pathogenesis have been developed (Table 4).

### 7.1. Nanoparticles with High Affinity to Aβ1–42 Peptide

This type of nanocarriers represents a completely new approach to treating AD. Such a method includes application of multifunctional nanoparticles modified by antitransferrin antibodies in combination with MAbs-IgG domains and transferrin ligands for delivery of the iAβ5 peptide, inhibiting the aggregates associated with AD [228]. Another strategy involves monoclonal antibodies (MAbs) targeting Aβ fibril formation [164]. For example, Kuo with coauthors investigated various modifications of gold nanoparticle systems loaded with the neuron growth factor in combination with transferrin, lactoferrin, p-aminophenyl-alpha-d-mannopyranoside and apolipoprotein E and added peptide sequence [218,219,220]. This sequence interacts with the transferrin receptor present in the microvascular endothelial cells of the blood–brain barrier, thus causing an increase in the permeability of the conjugate in brain, as shown by experiments in vitro and in vivo. It is highly relevant for the therapeutic applications of gold nanoparticles for molecular surgery in the treatment of neurodegenerative diseases such as AD. By irradiation with weak microwaves, Ab aggregates bound to the conjugate peptide sequence with a gold nanoparticle were destroyed by local dissipation of the absorbed energy in vitro [165]. Along with gold nanoparticles PLGA nanoparticles with surface functionalized by antitransferrin receptor monoclonal antibody (OX26) and anti-Aβ (DE2B4) have also been developed for the delivery of encapsulated iAβ5 into the brain. This system efficacy and toxicity were evaluated with porcine brain capillary endothelial cells (PBCECs) used as a BBB model [158].

Another form of nanocarriers [162] presents the construction of Aβ-targeted stealth liposomal nanoparticles using the Aβ-targeted lipid conjugate DSPE-PEG with Methoxy-XO4, which interconnect with amyloid. It was demonstrated that these nanoparticles selectively bind to amyloid plaques in the brain tissue of APP/PSEN1 transgenic mice. Ex vivo analyses of brain samples showed that injected nanoparticles efficiently bind both parenchymal plaques and cerebral amyloid angiopathy associated amyloid throughout the brain [162]. Along with other nanocarriers PEG coated and anti-Ab antibody-conjugated antioxidant (cerium oxide) nanoparticles were developed, and their effects on neuronal survival and BDNF signaling pathway were examined. For this nanoparticles protective effect against oxidative stress and Ab-mediated neurodegeneration were reported. In particular, these nanoforms specifically targets the Ab aggregates, produce an antioxidant effect and thereby perform neuroprotection [168].

β-sheet breaker peptides compose a class of compounds that are highly potent in Aβ1−42- or α-synuclein-inflicted cell toxicity neutralization. β-sheet breaker peptides are highly effective for Aβ amyloidogenesis inhibition as they are homologous to regions of the β-sheet hydrophobic carboxyl segments [229]. These peptides represent oligopeptides structurally analogous to specific epitopes of Aβ. Such substances have been also loaded in nanocarriers for Aβ-targeted delivery. It was demonstrated that the use of gold nanoparticles loaded with LPFFD peptide induces Aβ-aggregate inhibition, Aβ fibrils dissociation and Aβ-mediated peroxidase reduction [166]. TGN TGNYKALHPHNG and QSH D-enantiomeric peptide QSHYRHISPAQV were also conjugated to the surface of the nanoparticles for blood–brain barrier transport and Aβ42 targeting. Nanoparticles implication enhanced bioavailability of drug in the brain and produced neuroprotective effects in the mouse model of AD in vivo [159]. In another work [230] researchers developed a dual-functional nanoparticle drug delivery system loaded with β-sheet breaker peptide H102 (HKQLPFFEED). This peptide was encapsulated into liposomes to reduce the degradation level and was administrated nasally to increase brain penetration. It was demonstrated that the nanosystem application recovered spatial memory impairment in AD model rats both in a low dose and in a high dose [230].

Liposomes are frequently used for drug delivery during targeted AD therapy. For example, liposome nanoparticles are loaded with Aβ1–42 ligands with high affinity, phosphatidic acid and cardiolipin phospholipids [160]. Another research demonstrates the induction of amyloid degradation via administration of apolipoprotein E3-reconstituted high-density lipoprotein nanoparticles bond with Aβ oligomers [163]. Therapeutic effect was also reported for bifunctional liposomes, loaded with phosphatidic acid and peptide derived from apolipoprotein-E receptor-binding domain. In particular, peptide implication allowed penetrating through BBB and phosphatidic acid use caused Aβ aggregates degradation in APP/PS1 transgenic mice [161]. Thus, liposome administration reduced amyloid concentration and improved cognitive function in transgenic mice [161].

Nanoparticles using Congo Red/Rutin/MNP are another method of AD treatment. It was shown that the intranasal administration of Congo Red/Rutina-MNP contributed to the dissipation of amyloid plaques in MRI, increased drug delivery and reduced the production of active oxygen forms and Aβ-induced cytotoxicity in vitro. The intravenous administration of such nanosystems also increased the detection of amyloid aggregates, prevented memory deficit and neurological disorders in transgenic APPswe/PS1dE9 mice in vivo [231].

Multifunctional nanoparticles are being also used for diagnostic and therapy of other pathologies associated with Aβ accumulation. One of them is cerebral amyloid angiopathy described as a condition resulted from Aβ protein lodgment within the walls of cerebral vascular cells. The peptide accumulation leads to vascular inflammation and periodic hemorrhagic strokes. Therapeutic nano-transportable vehicles containing a polymeric core of chitosan-conjugated magnetivist (magnetic resonance imaging (MRI) contrast agent) have been developed for the treatment and early diagnosis cerebral amyloid angiopathy [157]. Cyclophosphamine, an agent that suppresses inflammation of brain vessels, was also loaded into the THB nanonucleus, and a Putreskin-modified F(ab’)2 fragment of an antiamyloid antibody, IgG4.1 (pF(ab’)24.1), was bound to the surface of the nanonucleus to target the cerebrovascular amyloid. Similar systems have already been tested in vitro with human endothelial cells (hCMEC/D3) and in vivo in laboratory mice [157]. It has been shown that other anti-inflammatory and antiamyloidogenic agents capable of reducing the inflammation of cerebrovascular vessels can also be loaded into such nanotransport systems [212].

### 7.2. Drugs Directed on the Inhibition of Tau Aggregation

Tau represents an intracellular protein, playing a crucial role in the microtubules stabilization. It was demonstrated that tau protein could respond to various factors via aggregates formation. The accumulation of these aggregates induces neurofibrillary tangles conglomeration in neurons and neurodegeneration. As the peptide contains two hexapeptide motifs in the repeat region, after pathological impact tau can randomly coil into the β-sheet structure and perform aggregation.

By now serious research has been done in the field of protein aggregation prevention in neurodegenerative diseases. In spite of investigation of various drugs from different classes like peptide inhibitors, natural substances, etc., little success was achieved. Some research has been performed in the area of nanoparticles, which offers a challenge in AD treatment improvement.

For example, a tau aggregation suppressor phenothiazine was loaded in polymer-based (PLGA-PEG) hydrophobic–glutathione coated nanoparticles [170]. Moreover, for some types of nanocarriers aggregation inhibitory activity were demonstrated of its own accord. The cadmium sulfide and iron oxide (Fe_3_O_4_) nanoparticles inhibit tau fibrillization in vitro. Despite its antiaggregation activity and excellent delivery properties, the in vivo experiments are hampered by interfering molecular species [171].

Red blood cell membrane-coated PLGA particles with T807 can also be used as a resource for tau aggregation inhibition. As T807 demonstrates high permeability for neurons and pronounced p-tau-specific tracing activity, it can be considered as a therapeutic agent of targeted delivery for AD treatment. One more virtue of that type of nanocarriers is red blood cell membranes, characterized by low immunogenicity and long half-life in the circulation, which allow performing effective drug delivery. Synergetic action of T807 with triphenylphosphine (TPP) produces an inhibitory effect on key components of AD pathogenesis associated with tau-phosphorylation and prevents neurodegeneration. Thus, the systemic administration red blood cell PLGA nanoparticles loaded with curcumin recover cognitive impairments in the mouse model of AD [232].

### 7.3. Acetylcholinesterase Inhibitors (AChE)

This class of drugs is characterized by many side effects due to their action on the peripheral tissues, high doses of administration and hepatic metabolism. These side effects are aggravated by the low pharmacokinetic (half-life) profiles of the drugs [150]. The loading of such preparation into nanocontainers would help to eliminate these side effects and to improve their bioavailability to the brain via an intranasal form.

One of the drugs used to treat AD is rivastigmine, which inhibits both the AChE and butyrylcholinesterase enzymes in brain and boosts the cholinergic function. It improves cognition and memory by increasing the ACh concentration in the brain. Nevertheless, the half-life period of rivastigmine in blood circulation is only 1.5 h, which complicates its therapeutic action. Loading drugs in nanocontainers increases its half-life and decreases the necessary dosage, thereby reducing side effects. For example, it has been already demonstrated for AChE inhibitors represented as naturally derived polyphenols, such as epigallocatechin-3-gallate and catechin placed into the sodium-taurocholate liposomal carrier [192], cholesterol/soy-lecithinated liposomal NP formulation [190], PLGA, PEG and gold NPs [233]. Yang and colleagues have also performed analysis of rivastigmine-liposome and CPP27 modified rivastigmine-loaded liposome for intranasal administration and showed the improvement of drug activity [191]. In particular, it was demonstrated that CPP modification increases the drug diffusion across the BBB by enhanced transcytosis and cell membrane permeability. It was acknowledged by the pharmacokinetic study indicated a significant increase of drug concentration in the various regions of the brain like cortex, hippocampus, and olfactory regional. Additionally, that modification has not exerted any toxic effects in the nasal cavity and in the brain. Improved drug properties were also demonstrated for rivastigmine via various nanoparticles. For example, rivastigmine-loaded mucoadhesive NP developed by Fazil et al., PLGA, PBCA NP by Joshi et al. and chitosan (polysorbate 80-coated) NP by Khemariya et al. showed an increased bioavailability [193,194,195]. In was also shown that gelling cationic nanostructured lipid carriers enhance brain distribution and pharmacodynamics [188]. Moreover, liposomes with RHN increased half-life, decreased the toxicity and improved memory and neurological deficits in Male albino rat with the model of AD [189].

Another AChE inhibitor using for AD treatment is galantamine. Some studies reported that galantamine could also reduce Aβ aggregation and neurotoxicity. Nevertheless, the application of galantamine is limited by its side effects such as nausea, vomiting, dizziness, drowsiness, loss of appetite, etc. That makes it inconvenient for the long-term oral treatment. In addition, galantamine is not able to cross the BBB efficiently [33]. For this reason, NP can improve response and bioavailability of galantamine. For example, Hanafy and colleagues constructed chitosan nanoparticle and cationic nanoparticles for intranasal drug administration [175]. Li et al. developed liposomal system, as a drug carrier for efficient delivery of galantamine through intranasal administration [173]. Other research groups created PLGA [174] and solid lipid NP [172] with galantamine, which have not only increased its bioavailability but also improved cognitive function and memory in Wistar rats.

Donepezil is another frequently used drug for AD treatment, represented as a centrally acting reversible AChE inhibitor. It is characterized by high pharmacokinetic properties with 100% bioavailable and a biological half-life of 70 h. However, the long-term administration leads to severe side effects, especially when taken in a high dose (>10 mg/day) [234]. The development of chitosan nanosuspension for intranasal administration [178] and the chitosan intranasal form of donepezil [235] promoted the usage of lower drug dose. Along with these nanoparticles LGA-b-PEG by Baysal et al. and liposomes by Al Asmari et al. were constructed for this drug [177,180].

Tacrine is AChE inhibitor acting as a histamine N-methyltransferase inhibitor. This drug produces various side effects including not only nausea, vomiting, dizziness, drowsiness, hallucination, confusion, insomnia and anxiety, but also hepatotoxicity, convulsion, urinary tract infection, bradycardia, depression, hypotension, etc. For the problem solution Qian and colleagues have developed in situ gelling systems for intranasal administration of tacrine [185]. That drug form enhanced the substance retention time in the nasal cavity and improved the nasomucosal absorbance. Wilson et al. have constructed a tacrine loaded chitosan nanoparticle coated with polysorbate 80 and demonstrated the improvement of its biodistribution [197]. Luppi et al. has also developed albumin nanoparticles carrying cyclodextrins for nasal delivery and drug properties advance [186].

Huperzine is another preparation used for AD treatment. For improvement of it bioavailability some research groups also constructed NP. For example, Yang et al. created nanostructured lipid carriers [181]. The treatment with huperzine lipid NP performed by Patel et al. demonstrated behavioral and memory improvement in the mouse model of AD [182].

### 7.4. Memantine, the NMDA Receptors Antagonist

Memantine hydrochloride (MEM) is the only anti-AD drug used in the US and Europe for the moderate-to-severe form of AD. The labeled formulation for memantine hydrochloride is Namenda, which is prescribed in tablet or capsule form. MEM prevents the aggregation of Aβ1–40 and enhances memory in rodents. Despite the best results in improving cognitive impairments in patients, the clinical applicability of MEM is limited. MEM should be administered daily, which leads to poor compliance with the drug regimen [205]. The development of nanocarriers with memantine can help to solve these problems. Toxicologically safe lipid nanoparticles with lipoyl-memantine have been elaborated and can be used to deliver memantine to the brain [206]. Gothwal et al. (2019) developed memantine-loaded polyamidoamine dendrimers. He used lactoferrin as a targeting ligand. The profile of memantine release from nanoparticles was slow and persisted for up to 48 h. The brain absorption of NPs was significantly higher than that of pure memantine. The in vivo research in an AlCl3-induced AD mice model demonstrated a considerable improving in behavioral tests [205].

### 7.5. Other Drugs for AD Treatment

Many literature data suggest that various growth factors may have therapeutic potential in the AD treatment. In particular, growth factors such as basic fibroblast growth factor, neurotrophins (nerve growth factors; glial-derived neurotrophic factor and BDNF and insulin-like growth factor (IGF-1/IGF-2)) and bone morphogenetic proteins are promising molecules [236]. In such a way PEG-PLGA nanoparticles loaded with basic fibroblast growth factor have been shown to improve cognitive function in the Morris test in rats exposed to intraventricular administration of β-amyloid 25–35 and ionic acid [221].

Metal chelators are another potential drugs to treat AD because they block the effects of metal ions stimulating the formation of Aβ aggregates and active oxygen forms. However, their use is limited due to non-specific interactions with metal ions and the inability to pass through BBB. Such problems have been solved by using controlled release systems of gold nanoparticles with mesoporous silica based on H_2_O_2_. Loading metal chelator CQ in these nanocontainers allowed targeted CQ release in the brain areas where the level of H_2_O_2_ is increased, for example, in Aβ-plaques [201].

Several recent studies have demonstrated that some antidiabetic drugs also have a therapeutic potential in AD patients. For example, Hsieh and colleagues has shown the inhibitory effect of gold nanoparticles on the fibrillogenesis of insulin fibrils [167]. Gold nanoparticles, coincubated with insulin, retarded the structural transformation of insulin fibrils into amyloid-like fibrils for about a week [167]. The neuroprotective effect was also demonstrated for nanoparticles with peptide NAPVSIPQ as a model drug and lactoferrin as a targeting ligand, which improved cognitive function in mice. As lactoferrin is a natural iron binding protein, whose receptor is highly expressed in both respiratory epithelial cells and neurons, its administration facilitates the nose-to-brain drug delivery of neuroprotection peptides like NAPVSIPQ [200].

### 7.6. Others Methods of AD Treatment

Currently, siRNA gene therapy is a promising method of treating of many brain disorders. Potentially it can also be used for the treatment of AD. An example of such a therapy method is the inhibition of the Beta-Secretase 1 (BACE1), which leads to suppression of disease development. Nanoparticles of the solid lipid with and without chitosan were used for directed delivery of BACE1 siRNA to the brain. These nanoparticles contained a RVG-9R peptide derived from the rabies virus glycoprotein, which enhanced the transcellular delivery of nanosystems to the brain during their intranasal injection. It was demonstrated that the investigated siRNA permeates the monolayer of the cell culture Caco-2 to a greater extent when released from nanocarriers, primarily from chitosan-coated solid lipid nanoparticles, than from bare siRNA [153].

The same research strategies were developed by other researchers. Wang and colleagues designed siRNA nanocarriers based on PEGylated poly(2-(*N*,*N*-dimethylamino) ethyl methacrylate) (PEG-PDMAEMA) modified with both the CGN peptide for BBB penetration and the Tet1 peptide for neuron-specific binding [154]. These nanocomplexes have been shown to successfully enter the cytoplasm of neurons, effectively suppressing the expression of BACE1 mRNA (by about 50%) and preventing synaptic damage caused by Aβ25-35. These results were also confirmed by studies of transgenic APP/PS1 mice, where the administration of nanoparticles stimulated neurogenesis in the hippocampus, inhibited the formation of amyloid plaques and tau-protein by repression of BACE1 mRNA. The administration of the nanocomplexes restored the cognitive function of the AD transgenic mice to the level of wild-type control without significant adverse effects on myelination [154].

Downregulation of the key enzyme in amyloid-β formation by delivering non-coding RNA plasmid is another potential way of AD treatment. It was shown that simultaneous delivery of the therapeutic peptide into the brain leads to the suppression of neurofibrillary tangles formation. In particular, a multifunctional nanocarrier contained the peptide to achieve the therapeutic gene, based on the PEGylated dendrigraphte of polylysines, has been developed for the treatment of AD [155]. Directional nanoparticle delivery has been investigated in transgenic AD mice [155].

Treatment of AD is extremely difficult because no precise mechanisms have been discovered for its development and pathogenesis. Despite that fact, a large number of different drug forms and therapy methods are being investigated. Among others, many nanocontainer systems have been constructed for different groups of drugs, inducing targeted delivery to pathological brain centers. Various clinical data provide an opportunity to hope that these strategies will bring their results.

## 8. Conclusions

The development of nanocontainer systems for the treatment of various diseases is a modern, promising and actively evolving field of science. Among other things, nanoparticles also contain agents for the treatment of neuropsychiatric diseases. All mental illnesses require long-term therapy, and patients need constant medication for months or even years. For this reason, it is important that the drug is delivered using nanocontainer systems that facilitate a longer release of the medicine. Various nanocontainers are opening up prospects for new forms of medication, such as subcutaneous transdermal drug delivery systems—“patches”. Non-compliance with therapy in patients is a very common and serious problem, which can be solved by prescribing injections of long-acting drugs or implants. Many neuroleptics have low bioavailability when used orally due to liver metabolism and low brain permeability, and drug delivery systems can improve these indicators. Prolonged therapy inevitably leads to side effects. Studies of some combined medications for the treatment of neuropsychiatric diseases have shown a reduction in side effects compared to pure drugs. Thus, the use of nanocontainer systems for the delivery of antidepressants, antipsychotics and drugs for the treatment of neurodegenerative diseases can contribute to more effective therapy and improve patients’ quality of life.

## Figures and Tables

**Figure 1 molecules-25-05294-f001:**
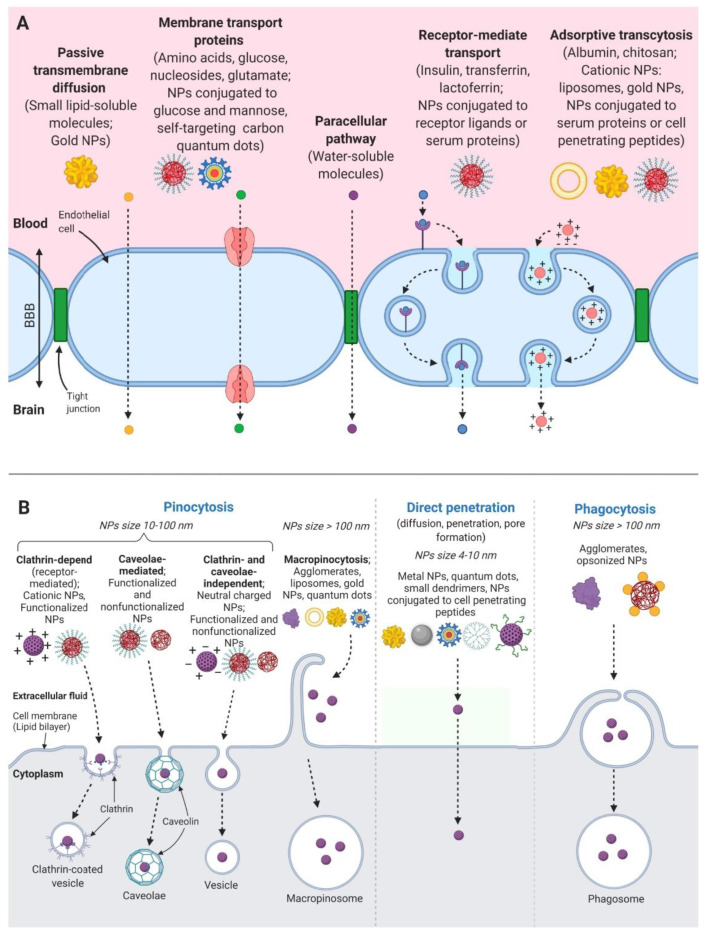
The main pathways of nanoparticles entering the brain. (**A**) The main mechanisms for blood brain barrier (BBB) penetration. (**B**) The mechanisms of nanoparticle transport across the cell membrane. BBB—blood-brain barrier; NPs—nanoparticles.

**Figure 2 molecules-25-05294-f002:**
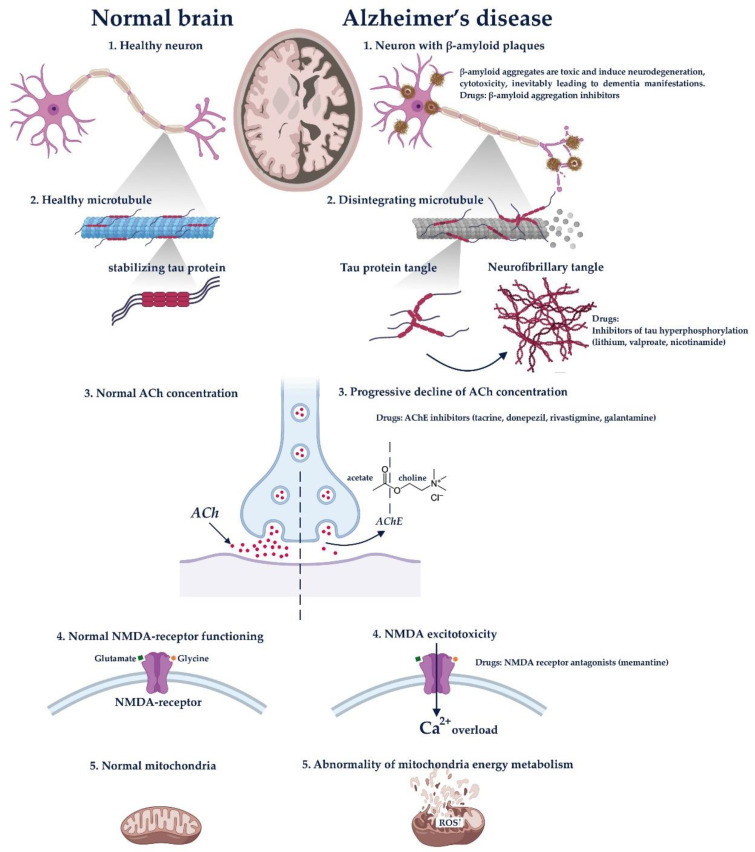
The main pathological pathway of Alzheimer’s disease and therapy methods, targeted to these mechanisms. Ach—acetylcholine, AChE—acetylcholine esterase, ROS—reactive oxygen species, NMDA—N-methyl-D-aspartate.

**Table 1 molecules-25-05294-t001:** Advantages of nanoparticles (NPs) administration (adm.) compared to other methods of drug delivery for CNS treatment.

Property	NPs	Intranasal Adm.	Transdermal Adm.	Oral Adm.	Intravenous Adm.
Controlled release	Available	Not available	Available	Not available	Not available
BBB penetration	Does not depend on the properties of drug Target ligands Transcytosis, endocytosis and exocytosis and others.	Does not depend on the properties of drug Direct transport	Depends on the properties of drug	Depends on the properties of drug	Depends on the properties of drug
Ability to targeted delivery	High	Not available	Not available	Not available	Not available
Bioavailability	High	High	Poor	Poor	Poor
Hepatic first-pass metabolism	Prevent	Prevent	Prevent	Not prevent	Not prevent
Toxicity	Less pronounced side effects because of lower concentrations of therapeutic drugs and more stable release characteristics.	Less pronounced side effects because of lower concentrations of therapeutic drugs and direct transport through BBB.	Less pronounced side effects because of lower concentrations of therapeutic drugs and more stable release characteristic	High chance and dose-depended side effects	High chance and dose-depended side effects

**Table 2 molecules-25-05294-t002:** Various categories of nanocarriers.

Nanocarrier	Description	Materials	BBB Penetration	Advantages	Disadvantages	References
**Polymer-Based Nanocarrier Systems**						
Polymeric NPs	The colloidal carriers are obtained from biodegradable and biocompatible natural or synthetic polymers into which drugs are loaded in either solid-state or solution.	Alginate, chitosan, gelatin, cellulose, polyacrylate, polyanhydride, PLGA, PACA, PCL, PEI, PLA, PEG.	The NPs are permeated the BBB through the tight junctions. The NPs are retained in the brain blood capillaries and are adsorbed of the NPs to the capillary walls. Endocytosis by the endothelial cells. Transcytosis. The inhibition of the efflux system by using polysorbate 80 as the coating agent. The surface functionalization with targeting ligands.	The significant potential for brain drug delivery across the BBB. Biocompatibility and biodegradability Good stability. Low cost. Less toxicity. Ease in production. Low immunogenic response. The mucoadhesive polymers (alginate, chitosan, cellulose, gelatin) can be used intranasally to bypass the BBB. Can be used as gelling/viscosity building agents to conquer nasal mucociliary clearance. Controlled drug release. The surface groups can be conjugated with targeting ligands.	Uncertain potential toxicity. Slow degradability.	[28,29,30,31,32,33]
Polymeric micelles	The core–shell structure from amphiphilic blocks copolymers that aggregate in aqueous solutions to form spheroidal NPs with a hydrophobic core and hydrophilic surface.	PEG, PLGA, cholesterol conjugated polyoxyethylene sorbitol oleate.	The surface functionalization with targeting ligands.	Biocompatibility. Possibility of solubilizing lipophilic drugs. The surface groups can be conjugated with targeting ligands.	The efficiency delivering across the intact BBB still needs further investigation.	[17]
Dendrimers	The monodisperse symmetric molecules that comprise a series of branching units around an inner core, which have a spheroidal shape and radially crowded layers.	Poly(amido amide) (PAMAM)	The internalization by brain capillary endothelial cells through a clathrin- and caveolane-mediated energy-depending endocytosis, also partly through macropinocytosis.	The core is loosely packed in comparison to the periphery and is suitable for the entrapment of drugs. The presence of numerous surface groups allows for high drug payload and multifunctionality. The surface groups can be conjugated with targeting ligands.	Potential toxicity.	[17,30,31,32]
**Lipid-Based Nanocarrier System**						
				Less or no toxicity, biodegradability, and ability to successfully deliver biomolecules, DNA, RNA, genes, antibodies, etc.		
Liposomes	The spherical vesicles with different sizes (20 nm to 100 μm), with an aqueous inner core enclosed by unilamellar or multilamellar phospholipid bilayers. Liposomes may have different surface charges and uni-, bi- or multi-lamellar structures.	Sphingomyelin, phosphatidyl choline, glycerophospholipids, cholesterol, phosphatidylcholine, cardiolipin.	Receptor-mediated and adsorption-mediated endocytosis (for small liposomes with a diameter not larger than 100 nm) The surface functionalization with targeting ligands Cationization of the conjugated ligands is another method to improve BBB transport rate of liposomes. Interactions of liposomes with cells can be realized by: adsorption, fusion, endocytosis, and lipid transfer.	Good biocompatibility. Non-toxic. Widely investigated. Encapsulate both hydrophilic and lipophilic drugs. Increase the solubility of drugs. Improve pharmacokinetic properties and protect the drugs from enzymatic degradation. Reduction of harmful side effects of drugs. Improving therapeutic effectiveness of drugs.	Opsonization and rapid clearance by macrophages of the mononuclear phagocytic system organs. This can be fixed by using surface coatings, such as PEG. Low drug transport rate. Poor stability. Difficulty of binding ligands to the surface as a result of the small number of available surface groups and steric hindrance.	[28,29,30,32,33]
Solid lipid NPs	Nano-sized dispersions of biocompatible lipids	The lipid core consists of triglycerides, diglycerides, monoglycerides, fatty acids, steroids, stearic acid or waxes stabilized by various surfactants	The surface functionalization with targeting ligands Can be used intranasally to bypass the BBB. The use of cationic lipids can improve mucoadhesion in the nasal cavity by promoting electrostatic interactions with mucus. Endocytosis by the endothelial cells. Adsorptive-mediated transcytosis of cationic NPs.	Good physical stability The capability to deliver lipophilic drugs dispersed in the hydrophobic core into cells. Improves bioavailability of drugs. Increases drug loading ability. Controlled drug release. Reduced cytotoxicity of drugs. Reduces the RES uptake, increases the retention and circulation time and protect drugs from degradation. The surface groups can be conjugated with targeting ligands.	Easy clearance by the reticuloendothelial system Low loading capacity Drug expulsion after crystallization Relatively high water content of the dispersions.	[29,30,33]
Nanoemulsions	Colloidal droplet system of oil-in-water (O/W) or water-in-oil (W/O) formulations stabilized with surface-active agents Oil droplet size of nanoemulsions ranges from 10 to 100 nm, making them interesting systems to improve drug delivery	Edible oils, such as flaxseed oil, pine-nut oil, hemp oil, fish oil as well as safflower oil and wheat-germ oil, biocompatible surfactants such as egg phosphatidylcholine which is one of the components of cell membrane lipids, deoxycholic acid, stearylamine, dioleoyltrimethyl ammonium propane.	The surface functionalization with targeting ligands Receptor-mediated endocytosis of cells	Biocompatibility. The ability to solubilize hydrophobic or hydrophilic drugs. Improves the bioavailability of the drug. The surface groups can be conjugated with targeting ligands.	Thermodynamic instability. Instability upon storage. Immediate drug release.	[17,32,33]
**Inorganic Nanoparticles**						
Mesoporous Silica NPs	Silica xerogels and mesoporous silica nanoparticles	*N*-Cetyltrimethylammonium bromide, Tetraethoxysilane	Transcytosis of vascular endothelial cells (PEGylated NPs)	Good biocompatibility. Stability. Highly porous framework. High drug loading efficiency. Controlled drug release.	Potential toxicity and adverse effect showed by recent studies Contribute to the level increase of reactive oxygen species Can induce the elevated production of malondialdehyde and decreased glutathione level	[30,34]
Gold NPs			Adsorption-mediated endocytosis Diffusion	Can easily penetrate cells because of their small size, high efficiency in cellular uptake. Easy conjugation to biomolecules. Can be easy functionalized by surface modification (small size, large surface area and pore volume, high reactivity).	Drug delivery efficiency needs further investigation.	[34]
Carbon nanotubes	Graphitic sheets rolled into single-walled or multiple walled tubes with an enormous surface area	Graphitic sheets	The surface functionalization with targeting ligands. PEG-modified carbon nanotubes also exhibited the capability of crossing the BBB.	Biocompatibility. Efficient loading of multiple molecules.	Potential toxicity. Drug delivery efficiency needs further investigation. Contribute to the level increase of reactive oxygen species. Similarity in carcinogenic potential between carbon nanotubes and asbestos. Can cause necrosis or apoptosis of macrophage cell lines and changes in cell morphology. Can induce proinflammatory response. Cannot be functionally integrated into the biological systems unless surface functionalization.	[30]
Iron oxide NPs	Iron oxide nanoparticles, which are also known as super paramagnetic nanoparticles	Iron oxide	Ability to cross the blood-brain barrier. Mechanisms of internalization and overall biodistribution are closely associated with their surface chemistry and hydrodynamic sizes.	Easy biodegradable (degraded iron can be absorbed by hemoglobin). Non-toxicity. Chemical stability in physiological conditions. Can be visualized by magnetic resonance imaging and it can be used for MR-imaging. Generate heat after exposure to an alternating magnetic field and can be used for magnetic hyperthermia and temperature-triggered drug release The possibility of using passive and active drug delivery strategies. Possibility of chemical modification by coating the iron oxide cores with various shells.	Tend to aggregate into larger clusters.	[17,30,31,34]

PLGA—polylactic acid-co-glycolic acid, PACA—poly (alkyl cyanoacrylate), PCL—polycaprolactone, PEI, polyethyleneimine, PLA—poly(lactic acid), PEG—poly(ethylene glycol), RES—reticular endothelial system.

**Table 3 molecules-25-05294-t003:** Summary of nanotechnology-based systems applied in the treatment of depression, anxiety and schizophrenia.

Drug	Nanocarrier	Components	PS, nm	ZP, mV	EE, %	LC, %	PDI	In Vivo Route of Administration	Outcomes	References
**Depression**										
Agomelatine	PLGA NPs	PLGA, PM407	116,06 ± 3523	−22.7	96	-	<0.3	i.n.	Prominent antidepressant activity	[58]
Agomelatine	Polymeric NPs	Propylene glycol, Plu F68, Ethanol, Transcutol HP.	107.64	−15	81.91	-	0.209	t.d.	Penetration enhancement of drug	[59]
Baicalein	Solid lipid NPs.	*N*-Acetyl Pro-Gly-Pro (to binding to neutrophil CXCR2 receptor), GMS, PM188, 1,2-dipalmitoyl-sn-glycero-3-phospho choline.	<100	−13.5, −12.6	98.7, 99.1	-	-	i.v.	The enhanced concentration of drug in the basolateral amygdala. Antidepressant effect in vitro and in vivo.	[60]
BDNF	Polyion complex formulation of BDNF (nano-BDNF)	Human recombinant BDNF, PEG-poly(l-glutamate) diblock copolymer	95	-	-	-	0.165	i.v.	Significant reduction in cerebral tissue loss. Improved memory/cognition, reduced post-stroke depressive phenotypes, and maintained myelin basic protein and brain BDNF levels.	[61]
HU-211 and curcumin	Solid lipid dual-drug NPs	Polyoxyethylene (40) stearate, stearic acid, and lecithin.	58.77 ± 1.7	−21.7 ± 0.4	-	0.74 ± 0.02, 18.34 ± 1.06	-	i.p.	Protection of PC12 cells from corticosterone-induced apoptosis. Antidepressant-like effect and enhanced fall latency in rotarod test, improved level of dopamine in the mice blood. NPs can deliver more curcumin to the brain and thus produce a significant increase in neurotransmitters level in brain tissue, especially in the hippocampus and striatum.	[62]
Curcumin and dexanabinol	Solid lipid NPs	Stearic acid, lecithin, polyoxyethylene stearate.	-	−22.6 ± 0.9	19.12 ± 1.43, 0.81 ± 0.04	-	-	-	NPs exerted antidepressant activity by targeting the endocannabinoid/CB1 receptor system.	[63]
Curcumin	Nanocapsules	Organic phase PCL, capric caprylic triglycerides, 106 sorbitan monoestearate, acetone	291–312	From −22 to −36	Close to 100%	-	-	via gavage	Antidepressant-like and antioxidant effects in a mouse model of Alzheimer’s disease. NPs were more effective than the free curcumin.	[64]
Duloxetine	Nanostructured lipid carriers	GMS, capryol 136 PGMC, Plu F-68, sodium taurocholate	80.17–127.73	-	-	-	-	i.n., i.v.	Better brain targeting efficiency than DLX solution when administered intranasally. Decreased side effects.	[65]
Folic acid	Niosomes	Different nonionic surfactants (Span 20, Span 60, Span 80, Tween 20, Tween 80, CL)	3050–5625	-	69.42	-	-	i.n.	Niosomes prepared with span 60 and CL in the ratio of 1:1 (50 mg: 50 mg) showed higher EE and better in vitro drug release of 64.2% at the end of 12 hrs and therefore considered as optimized formulation. About 48.15% of the drug was found to be absorbed through the nasal cavity at the end of 6 hrs.	[66]
Lithium carbonate	CS nanocomposites	CS, sodium tripolyphosphate anions, Tween 80.	90.68–220.81	+37.9	87 ± 1.21	28.87	-	p.o.	Reversed degenerative changes and gliosis in depression-induced animal models. Fortified targeted drug delivery and restrained adverse effects.	[67]
Minocycline hydrochloride (MH)	CS NPs	Tween 80, CS. Two types of NPs: MH loaded NPs, Tween 80 coated MH encapsulated NPs.	-	-	-	-	-	i.p.	NPs improved the therapeutic efficacy as well as safety of MH.	[68]
Selegiline hydrochloride	Thiolated CS NPs	CS, thioglycolic acid.	215 ± 34.71	+17.06	70 ± 2.71	-	0.214 ± 0.042	i.n.	Attenuation of the oxidative stress and restoring of the activity of the mitochondrial complex. Antidepressant-like effect in vivo	[69]
Silymarin	Nanostructured lipid carriers	Precirol solid lipid, Labrafac Lipophile oil.	519.00 ± 28.67	−12.95 ± 1.58	90.00 ± 3.20	-	0.66 ± 0.05	p.o.	Antidepressant-like effect, comparable with fluoxetine in mice. Significantly higher brain concentration by 12.46 fold superior to silymarin.	[70]
Thymoquinone (TQ)	Solid lipid NPs	Tween 80, GMS, PM188.	188.66 ± 8.94	−12.32 ± 1.04	68.60 ± 4.82	-	0.319 ± 0.04	p.o.	Higher amount of TQ reached the target region after administration. Higher levels of monoamines 5 hydroxytryptamine, dopamine and norepinephrine as compared to TQ suspension were demonstrated.	[71]
Tramadol HCl	CS NPs with mucoadhesive thermo-reversible gel.	CS, Thermo reversible mucoadhesive Plu-HPMC	152.0 ± 9.56	+31 ± 2.21	85 ± 3.23	-	0.143 ± 0.003	i.n.	Antidepressant-like effect in rat model of depression.	[72]
Trefoil factor 3 (TFF3)	cRGD-modified liposomes	Cyclic RGD (cRGD) peptide with high affinity for integrin receptors of leukocytes, soybean phosphatidylcholine, DSPE-PEG (PEG 2000), DSPG, CL	133	−21.8	27.6	-	-	i.p.	Brain targeted delivery in a murine model. Antidepressant-like effect of direct intra-basolateral amygdala administration of TFF3 solution in rats subjected to chronic mild stress.	[73]
Venlafaxine	CS NPs	CS glutamate.	167 ± 6.5	+23.83 ± 1.76	79.3 ± 2.6	32.25 ± 1.63	0.367 ± 0.045	i.n.	Enhanced uptake of venlafaxine to the brain.	[74]
Venlafaxine	PLGA NPs	PLGA, two ligands (Tf and TfRp) against transferrin receptor (TfR) to enhance access to brain across BBB.	206.3 ± 3.7	−25	48–50	10–12	0.041 ± 0.017	i.n.	Plain NPs demonstrated the highest ability to reach the brain vs. functionalized NPs.	[75]
Venlafaxine	Alginate NPs	-	-	-	-	-	-	i.n., i.v., p.o.	Improved antidepressant-like activity in comparison with the venlafaxine (i.n. and oral form). The greater brain/blood ratios for VNPs (i.n.)	[76]
Zn(^2+^)	PLGA NPs conjugated with glycopeptides	Antibodies against NCAM1 and CD44, PLGA conjugated with tetramethylrhodamine and glycopeptides.	190–210	From −0.5 to −10	-	-	-	-	NPs were able to cross the BBB and to deliver Zn(2+) ions at non-toxic concentrations. Easily modified for preferential targeting of specific cell populations.	[77]
**Anxiety Disorders**										
Peptide antisauvagine-30 (ASV-30)	Iron oxide NPs	Fe_2_O_3_, 3-aminopropyltriethoxysilane	5 ± 1	-	-	-	-	i.p.	Systemically administered NPs were observed in the brain. Association with neurons and reduced amphetamine withdrawal-induced anxiety in rats without affecting locomotion were demonstrated.	[78]
BUH	Thiolated CS NPs	CS, thiolated CS.	226.7 ± 2.52	-	81.13 ± 2.8	49.67 ± 5.5	-	i.n.	The brain concentration achieved after intranasal administration was significantly higher than after administration of BUH (i.v. and i.n.). Excellent bioadhesion.	[79]
BUH	Nanovesicles	Two types of nanovesicular in situ gels (P407-based and carbopol 974P-based). Span 20, Span 40, Span 60, CL, stearylamine, dicetylphosphate.	-	-	56.67–70.57	-	-	i.n.	Higher level of permeability for carbopol formulation in comparison to PM formulations. A 3.26 times increase of BUH bioavailability when loaded into the carbopol nanovesicular in situ gel in comparison to control (i.n.).	[80]
Diazepam	PLGA NPs	PLGA, PVA.	250	−23.3	66	-	-	-	In vitro drug release analysis showed sustained release of drug from nanoparticles. Correspondence to Korsmeyer–Peppas model.	[81]
Gallic acid (GA)	CS NPs	CS, Tween 80 (for coating (cGANP batch)). Two types of NPs: Ligand coated NPs (cGANP) and uncoated NPs (GANP).	103.33 ± 2.28	-	93.62	-	-	i.p.	Improved anxiolytic activity in mice. The plasma nitrite level decreased in GA, GANP and cGANP (most pronounced for cGANP) treated group as compared to saline treated control group while no change in plasma corticosterone levels was observed after any treatment.	[82]
**Schizophrenia and Bipolar Disorders**										
Aripiprazole (ARP)	Solid lipid NPs	Tristearin, Tween 80, sodium taurocholate.	85.05–2042	From -4.25 to −17.5	57.65–81.70	9.28–31.54	0.241–0.682	p.o.	Improved bioavailability of aripiprazole (1.6-fold compared to plain drug suspension). Enhancement of absorption and minimizing of the first-pass metabolism.	[83]
Aripiprazole	PCL NPs	PCL	199.2 ± 5.65	−21.4 ± 4.6	69.2 ± 2.34	-	-	i.n., i.v.	The enhanced brain targeting efficiency	[84]
Asenapine maleate	CS coated nanostructured lipid carrier	GMS, oleic acid, Tween 80, glycol CS.	184.2 ± 5.59	+18.83 ± 1.19	83.52 ± 2.59	-	-	i.n.	2.3 and 4 fold higher systemic and brain bioavailability compared to pure asenapine solution (i.n.).	[85]
Asenapine maleate	Transfersomes	SPC and sodium deoxycholate (SDC).	126.0	−43.7	54.96	-	0.232	t.d.	Significant increase of bioavailability after transdermal application compared with oral route.	[86]
Haloperidol	Lectin-functionalized, PEG–PLGA NPs	Mixture of Maleimide-PEG–PLGA and Me-PEG–PLGA, Solanum tuberosum lectin (STL).	132 ± 20	14.4 ± 0.1	73.2 ± 0.8	0.85 ± 0.01	0.07–0.22	i.n.	Increase of the brain tissue haloperidol concentrations by 1.5-3-fold compared to non-STL-NPs and other routes of administration.	[87]
Lithium carbonate	CS NPs	CS	162–419	+ 30	-	-	-	-	CS increased cellular uptake of lithium.	[88]
Lurasidone hydrochloride (LH)	Solid lipid NPs	GMS, PM188, sodium deoxycholate.	139.8 ± 5.5	−30.8 ± 3.5	79.10 ± 2.50	-	-	p.o.	Improved bioavailability of LH via lymphatic uptake along with improved therapeutic effect in MK-801 induced schizophrenia model in rats.	[89]
Lurasidone hydrochloride	Nanolipid carrier	Capryol 90, Tween 80, and Transcutol P.	207.4 ± 1.5	-	92.12 ± 1.0	-	0.392 ± 0.15	i.n.	2-fold increase of the drug concentration in the brain when compared with pure drug solution (i.n.)	[90]
Lurasidone	Mixed polymeric micelles	Plu F127, Gelucire 44/14.	175	-	97.8	-	-	i.n., i.v.	Improved brain distribution as well as kinetics of lurasidone via the intranasal route. The sustained release of lurasidone hydrochloride from micelles with better permeability and bioavailability.	[91]
Olanzapine	Solid lipid NPs	Two types of NPs with GMS and tripalmitin. Stearylamine, Tween 80.	165, 110	+66.50, +35.3	96.3, 67.2	-	0.34, 0.726	i.v.	23-fold increased bioavailability of olanzapine in brain and decreased clearance.	[92]
Olanzapine	Nanostructured lipid carriers	Mucoadhesive NPs was prepared by using carbopol 974P (MNLC (C)) and the combination of PM 407 and of HPMC K4M (MNLC (P + H)).	88.95 ± 1.7	−22.62 ± 1.9	88.94 ± 3.9	-	0,31 ± 0,01	i.n.	Nose-to-brain delivery of olanzapine MNLC (P + H) was considered as an effective and safe for CNS disorders.	[93]
Olanzapine	Micellar nanocarriers	Plu^®^ mixture of L121 and P123 in different ratios.	58.55 ± 2.47	-	75.03 ± 2.35	1.84 ± 0.06	0.27 ± 0.03	i.n., i.v.	The micelles (i.n.) demonstrated a conciliation between kinetic and thermodynamic stability, controlled drug-release characteristics and evoked minor histopathological changes in sheep nasal mucosa.	[94]
Olanzapine	Nanocapsules	Copolymer-functionalized PCL	254.9 ± 12.1	+22.2 ± 1.2	99.00 ± 0.05	-	0.03 ± 0.01	i.n.	Controlled release, improved retention of the drug on the nasal mucosa under continuous wash was demonstrated. Increased brain uptake and improved prepulse inhibition deficit induced by apomorphine in rats.	[95]
Olanzapine	CS NPs	CS. NPs were prepared with 20 or 60% loading.	208 ± 29 322 ± 18	-	86.7 ± 7.1, 87.6 ± 5.2	17.2 ± 1.4, 52.3 ± 3.1	-	i.n., i.v.	Olanzapine administered via intranasal CS NPs demonstrated the potential to improve the efficacy of systemic absorption thereby offering an efficient method of administration in noncompliant patients.	[96]
Olanzapine	PLGA NPs	PLGA, PM407, acetone, acetonitrile, tetrahydrofuran.	91.2 ± 5.2	−23.7 ± 2.1	68.91 ± 2.31	8.613 ± 0.288	0.120 ± 0.018	i.n., i.v.	6.35 and 10.86 times higher uptake than pure olanzapine solution (i.v. and i.n.) in vivo	[97]
Paliperidone	Nanolipomers	PCL as a polymeric core, Lipoid S75, and Gelucire^®^ 50/13 as a lipid shell and PVA as a stabilizing agent.	168.2 ± 0.7	−23.1	87.27 ± 0.098	-	0.22	p.o.	Sustained release up to 24 h and better ex vivo intestinal permeation for paliperidone compared to the corresponding polymeric, solid lipid NPs and drug suspension.	[98]
Paliperidone	in situ Gels	Carbopol 934, HPMC K4M, HP-β-CD in the form of inclusion complex of PLPD as nasal absorption enhancer.	-	-	-	-	-	i.n.	In vitro and ex vivo drug permeation, exhibited mucoadhesion and sustained drug release. The formulation containing HP-β-CD complex of paliperidone demonstrated higher level of drug permeation through sheep nasal mucosa without injury of nasal mucosa architecture.	[99]
Paliperidone palmitate	Micelles	d-alpha-tocopheryl polyethylene glycol 1000 succinate	26.5 ± 4.8	92.61 ± 2.5	-	-	-	i.m.	Improved antipsychotic effect and decreased adverse effects after micellar formulation application.	[100]
Paliperidone	Microemulsion	Muco-adhesive polymer, oleic acid	27.31 ± 1.86	−38.65 ± 2.39	-	-	0.241 ± 0.05	i.n., i.v.	Higher brain paliperidone concentrations compared to pure drug (i.v.)	[101]
Risperidone	Proteinoid NPs	Amino acids l-glutamic acid, l-phenylalanine, l-histidine, poly (l-lactic acid), PEG	86 ± 3	−16 ± 1	-	20 ± 0.1	-	i.v.	Enhanced antipsychotic activity compared to pure risperidone in mice	[102]
Risperidone	CS NPs	CS, tween 80, PM 188.	86	+36.6	77.96 ± 1.50	13–37	0.287	i.n.	Reduced stereotypical behavior score in experimental animals and reversed amphetamine effect.	[103]
Risperidone	Solid lipid NPs	Compritol 888 ATO, Plu F-127.	0.148 ± 0.028	−25.35 ± 0.45	59.65 ± 1.18	59.65 ± 1.18	0.148 ± 0.03	i.n., i.v.	Effective brain targeting in mice in vivo	[104]
Risperidone	Nanoemulsion	CS, capmul MCM, transcutol, propylene glycol	16.7 ± 1.21	−9.15 ± 2.14	98.86 ± 1.21	-	0.191 ± 0.04	i.n., i.v.	Effective delivery of significant amount of risperidone to the brain after intranasal administration.	[105]
Risperidone	Functionalized liposomes	Conventional liposomes consisting of SPC/CL, cationic liposomes containing SPC/CL/stearylamine, PEGylated liposomes consists of SPC/CL/distearylphosphatidylethanolamine-mPEG-2000	98.51 ± 6.82	−28.6 ± 3.62	58.86 ± 1.38	-	0.103–0.324	i.n., i.v.	Liposomal formulations provided enhanced bioavailability, less clearance rate, higher mean residential time and better response in vivo compared to conventional formulations.	[106]
Risperidone	CS lipid NPs	CS	132.7	-	-	7.6	-	i.n., i.v.	Increased nose to brain drug delivery compared to pure drug suspension of equivalent dose.	[107]
Quetiapine fumarate	Solid lipid NPs in situ gel	Heat-melting GMS, Span 80, butanol, PM407, PM188.	307.1 ± 17.7	+ 57.2 ± 0.24	97.6 ± 0.58	-	-	i.n., i.v., p.o.	Stable and effective brain delivery, amelioration of the damages induced by MK-801 in rat model of schizophrenia.	[108]
Quetiapine fumarate	Nanoemulsion system	HLBs of Emalex LWIS 10, PEG 400, Transcutol P, Capmul MCM, Tween 80.	144 ± 0.5	−8.131 ± 1.8	-	-	0.193 ± 0.04	i.n.	2-fold increase of the drug release compared to pure drug. Better direct nose-to-brain drug transport.	[109]
Quetiapine Fumarate	Liposome	Egg phosphatidylcholine, CL	139.6	−32.1	75.63 ± 3.77	-	-	i.n.	Higher level of diffusion. Better potential to deliver drugs to the brain than by the pure solution.	[110]
Quetiapine fumarate	Microemulsion with and without CS	CS, methyl-β-cyclodextrin, Capmul MCM EP, labrasol, Tween 80, Transcutol-P.	35.31 ± 1.71	20.29 ± 1.23	-	-	0.249 ± 0.03	i.n., i.v.	Enhanced brain uptake of quetiapine and improved bioavailability.	[111]
Quetiapine fumarate	CS NPs	CS, sodium tripolyphosphate	131.08 ± 7.45	+ 34.4 ± 1.87	89.93 ± 3.85	-	0.252 ± 0.064	i.n., i.v.	Significantly higher brain/blood ratio and 2 folds higher nasal bioavailability in the brain in comparison to pure drug solution (i.n.).	[112]

BBB—Blood–brain barrier; NPs—nanoparticles; PS—particle size; ZP—zeta potential; EE—drug entrapment efficiency; LC—loading capacity; PDI—polydispersity index; i.m. —Intramuscular route of administration; i.n.—intranasal route of administration; i.p.—intraperitoneal route of administration; i.v.—intravenous route of administration; p.o. —per oral route of administration; t.d.—transdermal route of administration; BDNF—brain-derived neurotrophic factor; BUH—Buspirone hydrochloride; CL—cholesterol; CS—chitosan; GMS—glyceryl monostearate; HPMC—hydroxypropyl methyl cellulose K4M; PCL—poly(ε-caprolactone); PEG—(poly(ethylene glycol); PLGA—poly(lactic-co-glycolic acid); Plu—pluronic; PM—poloxamer; PVA—polyvinyl alcohol; SPC—soya-phosphatidylcholine.

**Table 4 molecules-25-05294-t004:** Summary of nanotechnology-based systems applied in the treatment of Alzheimer’s disease.

Drug	Nanocarrier	Targeting Ligand	Components	PS, nm	ZP, mV	EE, %	LC, %	PDI	In Vivo Route of Administration	Outcomes	References
**Amyloid—Related Nanoparticles**											
Pioglitazone	Nano lipid carriers	-	Tripalmitin, MCM, stearyl amine	211.4 ± 3.54	+14.9 ± 1.09	70.18 ± 4.5	-	0.257 ± 0.108	i.n.	The NC significantly improved the nasal permeability of pioglitazone ex-vivo.	[152]
BACE1 siRNA	Solid lipid NPs	Peptide derived from rabies virus glycoprotein (RVG-9R)	CS-coated and uncoated NPs	419.47 ± 24.36, 469.71 ± 49.07	−12.52 ± 0.99, +14.47 ± 0.19	-	-	0.26 ± 0.09, 0.30 ± 0.04	i.n.	The siRNA permeated the monolayer (Caco-2) to a greater extent when released from any of the studied formulations than from bare siRNA, and primarily from CS-coated NPs.	[153]
siRNA against BACE1	Nano complexes CT/siRNA, composed of CGN-PEG-PDMAEMA and Tet1-PEG-PDMAEMA (1:1)	CGN peptide, Tet1 peptide	PEGylated poly(2-(*N*,*N*-dimethylamino) ethyl methacrylate) (PEG-PDMAEMA)	70–80	+10	-	-	-	i.v.	The NC entered the cytoplasm of the neuron cells, inducing effective gene silence (about 50% decrease in BACE1 mRNA levels) and reversing synaptic injury. The NC significantly decreased BACE1 mRNA and the amyloid plaques, suppressed phosphorylated tau protein levels, and promoted hippocampal neurogenesis. NC restored the cognitive performance of the AD transgenic mice to the level of wild-type control.	[154]
Plasmid DNA encoding BACE1-AS siRNA, d-peptide	PEGylated DGL NPs	Peptide from rabies virus glycoprotein (RVG29)	Dendrigraft poly-l-lysines 10 (DGL), α-Malemidyl-ω-*N*-hydroxysuccinimidylpolyethyleneglycol.	110	+ 7.72 ± 2.80	-	-	0.3	i.v.	Successful codelivery of drugs crossed the BBB. Simultaneous delivery of the therapeutic peptide into brain led to the reduction of neurofibrillary tangles. The memory loss rescue in AD mice was also observed.	[155]
BACE1 siRNA	Dendrimer NPs	Apolipoprotein A-I (ApoA-I), NL4 peptide.	Dendrigraft poly-l-lysines, α-Maleimide-ω-*N*-hydroxysuccinimidylPEG	79.26	+3.53	97.05	-	0.216	-	The NC effectively targeted both BBB and PC12 cells and down-regulated BACE1 gene expression in PC12 cells.	[156]
Cyclophosphamide	Theranostic nanovehicles	Putrescine modified F(ab′)_2_ fragment of antiamyloid antibody, IgG4	CS, MRI contrast agent, pentasodiumtripolyphosphate,	239 ± 4.1	11.9 ± 0.5	-	21,7 ± 1,31	-	i.v.	NPs successfully targeted cerebrovascular amyloid.	[157]
Peptide iA5	PLGA NPs	Anti-transferrin receptor monoclonal antibody (OX26), anti-A (DE2B4)	PLGA	166 ± 2	−13 ± 1	63 ± 9	-	0.10 ± 0.04	-	The uptake of NPs with a controlled delivery of the peptide iA5 was substantially increased compared to the NPs without monoclonal antibody functionalization.	[158]
Peptide H102	PEG-PLA NPs	TGN and QSH peptides.	PEG-PLA	125.5.10 ± 2.26	−29.33 ± 0.15	58.49 ± 0.86	0.54	0.127. ± 0.010	i.v.	This dual-functional drug delivery system effectively increased the H102 accumulation at brain Aβ42 concentrated in the hippocampal region and provided better neuroprotective effects in the AD model mice.	[159]
-	Liposomes and solid lipid NPs	Phosphatidicacid, cardiolipin	Monosialogangliosides, disialogangliosides, trisialoganglioside, sphingomyelin, CL, diphosphatidylglycerol, 1-palmitoyl-oleoyl-PC, phosphatidylethanolamine, Sephadex G75.	145, 76	−37.89, −43.30	-	-	-	-	NPs with surfacephosphatidic acid and cardiolipin demonstrated pronounced affinity to Aβ1e42 fibrils	[160]
-	Liposomes	Peptide from the apolipoprotein-E receptor- binding domain, phosphatedic acid.	Dimyristoyl-PA, sphingomyelin, CL.	121 ± 7	−18.7 ± 4	-	-	0.15	i.p.	Decrease of the total brain-insoluble amyloid peptide was demonstrated. Amelioration of mice impaired memory was observed.	[161]
-	Liposomes	MethoxyXO4	DSPE-PEG, CL, DPPC.	150	-	-	-	-	i.v.	The NPs appeared to cross the BBB and to bind with Aβ plaque deposits, marketing the parenchymal amyloid deposits and vascular amyloid.	[162]
-	Apolipoprotein E3 NPs	High density lipoprotein	Lipid free ApoE3, DMPC.	27.9 ± 8.9	−4.07 ± 0.83	-	-	0.30	i.v.	Four-week daily treatment with NPs decreased Aβ deposition, attenuated microgliosis, ameliorated neurologic changes, and rescued memory deficits in an AD animal model.	[163]
-	Monodisperse iron Oxide NPs	Monoclonal antibody against fibrillar human amyloid-β	-	8	-	-	-	-	i.v.	The targeting ability of NPs to cerebrovascular amyloid was demonstrated.	[164]
-	Gold NPs	peptide CLPFFD, peptide sequence THRPPMWSPVWP	Gold (III) chloride hydrate.	13 ± 1.7	−41 ± 2	-	-	-	i.p.	Increase of the permeability of the conjugate into the brain was shown.	[165]
-	Gold NPs	-	Gold NPs, polyoxometalate with Wells–Dawson structure, Aβ1-40 peptide	21.7	−36.8	-	-	-	i.v.	NPs inhibited Aβ aggregation, dissociated Aβ fibrils and decreased Aβ -mediated peroxidase activity and Aβ -induced cytotoxicity.	[166]
-	Gold NPs	-	Gold colloid solution	10 ± 2	-	-	-	-	-	NPs disrupted insulin amyloid fibrillation resulting in construction of fibrils that are shorter and more compact.	[167]
-	PEG-coated cerium oxide NPs	Aβ antibody	PEG	-	−37	-	-	-	-	The rescue of neuronal survival was demonstrated.	[168]
**Tau–Related Nanoparticles**											
Nicotin-amide	Solid lipid NPs	-	Stearic acid, phospholipon^®^ 90G, sodium taurocholate. Three types of particles functionalized by phosphatidic acid, polysorbate 80, phosphatidylserine.	124 ± 0.8	−46.1 ± 0.65	41.3 ± 0.41	-	-	i.p., i.v.	The phosphatidylserine-functionalized NPs improved the cognition, preserving the neuronal cells and reducing tau hyperphosphorylation in a rat model of AD.	[169]
Methylene blue	PLGA NPs	-	PLGA-b-PEG	136.5 ± 4.4	-	-	-	-	-	The reduction of both endogenous and over expressed tau protein levels in human neuroblastoma SHSY-5Y and HeLa cells was shown.	[170]
-	Protein-capped metal NPs	-	Two types of NPs: iron oxide (Fe_3_O_4_) NPsare capped with hydrolytic proteins from fungi, cadmium sulfide (CdS) NPs are capped with the mixture of four different proteins.	10–20	-	-	-	-	-	CdS NPs demonstrated dual properties of inhibition and disaggregation of Tau.	[171]
**Acetylcholinesterase Inhibitors**											
Gal	Solid lipid NPs	-	Plu F127, Tween 80, glyceryl behenate.	92.0 ± 3.51	−17.22 ± 1.1	83.42 ± 0.63	-	0.380 ± 0.16	p.o.	NPs restored memory in cognitive deficit rats. Increased bioavailability compared to the plain drug was demonstrated.	[172]
Gal	Flexible liposomes	-	Propylene glycol, soya PC, CL	112 ± 8	−49.2 ± 0.7	83.6 ± 1.8	-	-	i.n., p.o.	The efficiency of acetylcholinesterase inhibition was greatly enhanced by i.n. administration compared with p.o. administration, especially after loading of the drug in flexible liposomes.	[173]
Gal	Nano-emulsions	-	PLGA	21.5 ± 0.25	−11.18 ± 0.89	98.47 ± 0.43	56.87 ± 3.48	-	-	Non-cytotoxic drug-loaded NPs have been obtained with high encapsulation efficiencies and sustained drug release, maintaining drug pharmacological activity.	[174]
Gal	CS complex NPs	-	CS, Tween 80, sodium tripolyphosphate	-	-	-	-	-	i.n.	NPs exhibited a significant decrease of AChE protein level and activity in rat brains.	[175]
Gal	CS NPs	-	CS, sodium tripolyphosphate.	190 ± 1.159	+ 31.6 ± 9.75	23.34	9.86	0.276 ± 0.006	i.n.	NPs were detected in the olfactory bulb, hippocampus, orbitofrontal and parietal cortices 1 h after i.n. administration.	[176]
Donepezil	Liposomes	-	CL, PEG, DSPC.	102 ± 3.3	−28.31 ± 0.85	84.91 ± 3.31	-	0.28 ± 0.03	i.n.	The bioavailability of drug in the plasma and in the brain increased significantly. The formulation was safe and free from toxicity	[177]
Donepezil	Nano suspension	-	CS, tripolyphosphate.	150–200	-	92–96	40–48	0.341	i.n.	No mortality, hematological changes, body weight variations and toxicity in animals were observed.	[178]
Donepezil	Mango gum polymeric NPs	-	CS, mango gum, Span 80. NPs were prepared by two methods.	135. 55, 95.1 2	+ 8.2 5, + 25.9 2	72 ± 3.62, 85 ± 2.14	51 ± 5.42, 76 ± 2.26	0.53, 0.26	i.v.	The brain targeting was achieved.	[179]
Donepezil	PLGA-b-PEG nanoparticles	-	PLGA, PEG, Plu F68.	174-240	From −11.32 to −20.49	52–61	-	0.20–0.36	-	NPs crossed the BBB and showed a controlled release profile in this system. NPs administration caused a significant dose-dependent decrease in both gene and protein expression levels of IL-1b, IL-6, GM-CSF and TNF-a.	[180]
Hup	Nano structured lipid carriers	-	Cetyl Palmitate, Miglyol^®^812, soybean PC, Solutol HS15^®^	120	−22.93 ± 0.91	89.180.28	1.460.05	-	-	A burst release at the initial stage and followed by a prolonged release of drug from NPs was up to 96 h.	[181]
Hup	Micro emulsion, solid lipid NPs, nano-structured lipid carriers.	-	Glyceryl monostearate, glyceryl mono-dicaprylate, TranscutolP, Tween 80, triethanolamine.	119–148	From −4 to −20	60.05 ± 5.84	-	0.3–0.422	t.d.	In vitro permeation profiles in rat skin exhibited zero-order kinetics. The significant improvement in cognitive function in mice was observed.	[182]
Hup	Nanoemulsion	Lf	Isopropyl myristate, Capryol 90, Cremophor EL, Labrasol.	15.24 ± 0.67	−4.48 ± 0.97	-	-	0.128 ± 0.025	i.n.	Intranasal Lf-nanoemulsion significantly enhanced drug delivery to the brain compared to pure nanoemulsion.	[183]
Hup	Mucoadhesive and targeted PLGA NPs	Lf	N-trimethylated CS, PLGA 5050 2A.	153.2 ± 13.7	+35.6 ± 5.2	73.8 ± 5.7	-	0.229 ± 0.078	i.n.	The NPs facilitated the distribution of drug in the brain.	[184]
Tacrine	In situ gels	-	Plu F127, Plu F68, CS, PEG 8000	-	-	-	-	-	i.n.	The enhanced nasal residence time, improved bioavailability, increased brain uptake of the drug and decreased exposure of metabolites were shown.	[185]
Tacrine	Albumin NPs carrying beta cyclodextrin and two beta cyclodextrin derivatives	-	Beta cyclodextrinderivatives, hydroxypropyl beta cyclodextrin, sulphobutylether beta cyclodextrin.	177–266	From −10 to −10.9	85–91	12.5–22.0	0.228–0.327	i.n.	The presence of the different beta cyclodextrins affected drug loading and differently modulated NPs mucoadhesiveness and drug permeation.	[186]
Tacrine	CS NPs	-	CS, Polysorbate 80	41 ± 7	+34.7 ± 1.5	77–83	10,86 ± 0,30	-	i.v.	The NPs coated with 1% Polysorbate 80 altered the biodistribution pattern of NPs.	[187]
Riv	Nano structured lipid carriers	-	Glyceryl monosterate, Capmul MCM C8, lecithin, stearylamine, Tween 80	123.2 ± 2.3	32 ± 1.2	68.3 ± 3.4	-	-	i.n., i.v.	The faster regain of memory loss in amnesic mice with 5-fold decrease in escape latency with NC compared to plain rivastigmine solution was shown.	[188]
Riv	Liposomes	-	1,2-Diacyl-sn-glycero-3-phosphocholine, dihexadecyl phosphate, CL	67,51–528.7	From −6.6 to −25.1	75–97	-	0.612–0.755	Subcutaneously	NPs resulted in faster memory regain and amelioration of metabolic disturbances in AD rats.	[189]
Riv	Liposomes	-	Soya lecithin, CL	10.0 ± 2.8 µm	-	80.0 ± 5.0	-	-	i.n., p.o.	Intranasal liposomes demonstrated a longer half-life in the brain than intranasally or orally administered pure drug.	[190]
Riv	Liposomes	Cell penetrating peptide	DSPE-PEG, CL, egg PC	178.9 ± 11.7	−8.6 ± 2.4	30.5 ± 8.0	-	0.333 ± 0.032	i.n., i.v.	i.n. administration demonstrated the capacity to improve rivastigmine distribution and adequate retention in CNS regions compared to i.v. administration.	[191]
Riv	Liposomes	-	CL, DPPC, methyl cellulose, dimethyl-β-CD, sodium taurocholate	3.37 ± 0.00 µm	−4.30 ± 0.66	25,2	-	-	p.o., i.p.	The highest acetylcholinesterase inhibition was observed for rivastigmine-sodium-taurocholate liposomes.	[192]
Riv	PLGA NPs, PBCA NPs	-	PLGA, PBCA	135.6 ± 4.2, 146.8 ± 2.7	−23.7 ± 1.18, −3.9 ± 0.58 m	74.46 ± 0.76, 57.32 ± 0.91	-	-	i.v.	The faster regain of memory loss in amnesic mice with both PLGA and PBCA NPs compared to rivastigmine solution was demonstrated.	[193]
Riv	CS NPs	-	CS, Polysorbate 80	45,16 ± 1,56	35.08 ± 1. 5	83.26 ± 3.1	43.48 ± 1.3 0	-	i.v.	The Polysorbate 80 Coated CS NPs of rivastigmine was formulated and evaluated for brain delivery.	[194]
Riv	CS NPs	-	CS	185.4 ± 8.4	38.4 ± 2.85	85.3 ± 3.5	43.37 ± 3.9	0.391 ± 0.065	i.n., i.v.	NPs demonstrated better brain targeting efficiency and represented a promising approach for i.n. delivery of rivastigmine for the treatment and prevention of AD.	[195]
**Others**											
p38 MAPK inhibitor PH797804	Nanoemulsion-based CS nanocapsules	-	Span 85, oleic acid, Tween 20, CS.	406.1	−18.5 ± 2.9	21.5 ± 2.7	3.5 ± 0.5	0.287	i.n.	The p38 MAPK inhibitor encapsulated in CS nanocapsules reduces p38 MAPK activity in brain cells in the cerebral cortex and hippocampus to a lower extent.	[196]
Erythropoie-tin	Solid lipid NPs	-	Glycerin monostearate, Span 80, Span 60, dichloromethane, Tween 80	219.9 ± 15.6	−22.4 ± 0.8	-	-	(0.187 ± 0.03	i.p.	NPs reduced the oxidative stress, and beta-amyloid plaque deposition in the hippocampus more effectively than the pure drug. The memory was significantly restored in cognitive deficit rats treated with NPs.	[197]
Piperine	Monoolein cubosomes	-	Peceol^®,^ Tween 80, poloxamer, Cremophor.	167.00 ± 10.49	−34.60 ± 0.47	86.67 ± 0.62	-	0.18 ± 0.01	p.o.	The NPs significantly enhanced drug cognitive effect and even restored cognitive function to the normal level.	[198]
NAP	High density lipoprotein nanostructure	Mono sialotetrahexosyl ganglioside (GM1)	apoE3-reconstituted high density lipoprotein (rHDL), 1,2-dimyristoyl-sn-glycero3-phosphocholine (DMPC) liposome	25.42 ± 1.18	−15.70 ± 0.93	64.39 ± 12.84	-	0.218	i.n.	Protection of neurons from cell toxicity, a reduction of Aβ deposition and a rescue of memory in AD model mice were demonstrated.	[199]
NAP	PEG-PCL NPs	Lf	PEG-PCL, coumarin-6.	88.4 ± 7.8	23.56 ± 0.96	47.61 ± 2.36	0.62 ± 0.013	0.22 ± 0.033	i.n.	The neuroprotective and memory improvement effect of Lf-NP was observed. These results were also confirmed by the evaluation of acetylcholinesterase, choline acetyltransferase activity and neuronal degeneration in the mice hippocampus.	[200]
Metal chelatorclio-quinol	Gold NP-capped mesoporous silica	-	Mesoporous silica NPs (*N*-Cetyl trimethylammonium bromide, tetraethoxysilane)	52.13	+26.7	-	-	-	i.v.	The NPs reduced the cell membrane disruption, microtubular defects and apoptosis.	[201]
Selenium, sialic acid	Sialic acid modified selenium NPs	Peptide-B6	Sodiumselenite,	95	−14.4	-	-	-	-	The high permeability across the BBB was shown. The effectively inhibition Aβ aggregation was demonstrated. Therefore, NC did not disaggregate preformed Aβ fibrils into non-toxic amorphous oligomers.	[202]
Resveratrol, grape extract	Solid lipid NPs	Antitransferrin receptor mono clonal antibody (OX26 mAb)	Cetylpalmitate, polysorbate 80.	254 ± 17	−4.0 ± 0.1	92 ± 7, 95 ± 2	-	0.23 ± 0.05	-	The cellular uptake of the OX26 NPs was substantially more efficient than that of normal NPs and NPs functionalized with an unspecific antibody.	[203]
Resveratrol (Res)	Delivery system MSe-Res/Fc-β-CD/Bor	-	Mesoporous nano-selenium (MSe), β-CD, borneolBor)	160	-	>70	-	-	i.v.	NC inhibited aggregation of Aβ, mitigated oxidative stress, suppressed tau hyperphosphorylation and improving memory impairment in AD mice.	[204]
Memantine	Polyamido amine (PAMAM) dendrimers	Lf	PAMAM	131.72 ± 4.73	+ 20.13 ± 0.94	71.1 ± 4.84	-	0.16 ± 0.025	i.v.	A significant improvement in behavioral responses was demonstrated.	[205]
Lipoyl–memantine	Solid lipid NPs	-	Stearic acid	170	−33.8	88	-	0.072	Through the gastrointestinal tract	The NPs demonstrated low toxicity.	[206]
Amlodipine	Diamond NPs	-	Nanodiamond powder	31.1 ± 8.2	-	41	-	-	-	The highest percentage of loaded amlodipine onto nanodiamond particles was achieved in alkaline medium using 2 mMNaOH at a corresponding pH of 8.5.	[207]
Curcumin	Low density lipoprotein mimic nanostructured lipid carrier	Lf	Carboxylated PEG (100) monostearate, CL, glycerol trioleate	103.8 ± 0.6	−5.80 ± 0.73	96.51 ± 1.87	2.60 ± 0.17	0.15 ± 0.022	i.v.	The Lf NPs could effectively permeate BBB and preferentially accumulate in the brain. Superior efficacy of Lf NPs in controlling the damage associated with AD.	[208]
Curcumin	RBC membrane camouflaged HSANPs	T807, tri phenyl phosphine (TPP)	Amino-T807, carboxy-TPP, PEG, red blood cell (RBC) membrane, human serum albumin (HAS).	<120	-	88.43 ± 1.25	4.87 ± 1.04	-	i.v.	The NPs relieved AD symptoms by mitigating mitochondrial oxidative stress and suppressing neuronal death.	[209]
Curcumin, NGF	Liposomes	Wheat germ agglutinin (WGA), cardiolipin (CL)	CL, soybean PC, 1,2-Dipalmitoylsn-glycero-3-phosphocholine.	122–142	−5.2 to −18.3	20.5–56.7	-	-	i.v.	The NPs inhibited the expression of phosphorylated p38, prevented neurodegeneration of cells, enhanced the quantities of p-neurotrophic tyrosine kinase receptor type 1 and p-extracellular signal-regulated kinase 5.	[210]
Curcumin, selenium	PLGA nanospheres	-	PLGA, selenium NPs	160 ± 5	-	-	11.5	Low	i.v.	The NPs provided the enhanced therapeutic efficacy in AD lesions.	[211]
Curcumin or dexamethasone	Polymeric nanocore	Antiamyloid antibody IgG4.1	CS, gadolinium-diethylene triamine pentaacetic acid, hydroxypropyl-beta-cyclodextran, Magnevist^®^, 125I	145 ± 5,4, 157,6 ± 3,4	7,7 ± 0,4, 4,5 ± 0,5	-	-	-	i.v.	The NPs targeted cerebrovascular amyloid deposits selectively.	[212]
Andrographolide	Human albumin NPs	-	Human albumin	210.4 ± 3.2	−20.3 ± 1.5	99.1 ± 0.2	-	0.10 ± 0.01	i.v., i.p.	In the step-down inhibitory avoidance test, NPs improved mice performance. The presence of NPs both in the pE3-Aβ plaque surroundings and inside the pE3-Aβ plaque was observed. The anti-inflammatory activity was shown.	[213]
Flurbiprofen	Dendrimer NPs	-	Phenylalanine	-	-	-	-	-	-	Efficiency of NC influence on the γ-secretase enzyme in target cells was shown. Eventual drug release by hydrolysis of the carrier was demonstrated.	[214]
Tarenflurbil (TFB)	Solid lipid NPs, PLGA NPs	-	Monostearate, stearic acid, soya lecithin, Tween 20, PLGA, DMAB, PF-68, PVA	169.87 ± 10.98, 133.13 ± 7.82	−23.13 ± 2.32, −30.25 ± 2.11	57.81 ± 5.32, 64.11 ± 2.21	−10	0.24 ± 0.04, 0.21 ± 0.02	i.n.	The absolute bioavailability of the NPs was higher than TFB solution suspension.	[215]
Metformin	Nano liposomes	-	Phosphatidylserine.	148.3 ± 5.6	−34.8 ± 2.8	37 ± 2.3	-	<0.3	i.p.	The learning and memory parameters significantly improved in AD-rats treated with NPs. The decreased cytokine levels of IL1-β, TNF-α and TGF-β in hippocampal tissues were shown. A reduction in inflammatory and necrotic neural cells, and an increase neurogenesis was demonstrated.	[216]
Saxagliptin	CS-l-valine based NPs	l-valin	CS, BocL-valine	385 ± 21	+ 0.554 ± 0.110	56.23 ± 13.44	-	0.574 ± 0.125	i.p.	The NPs were highly stable in the plasma releasing only a minute after administration of the drug. Pronounced accumulation of drug from the NPs was demonstrated. The pure substance showed no detectable amount of the drug after 24 h.	[217]
NGF	Liposomes	Lf	CL, DSPE-PEG, DPPC, PEPEG	About 110	From −3 to −11	-	-	-	-	Lf/NGF-liposomes comprising CL and DPPC were physically stable with high biocompatibility to HBMECs and HAs cells.	[218]
NGF	Liposomes	P-aminophenyl-a D manno-pyrano-side, apolipo protein E	Soybean PC, phosphatidic acid, DPPC, cardiolipin, DSPE-PEG, CL	<170	From −4 to −10	35.1 ± 3.2	-	-	-	NPs were capable to enhance the NGF delivery across the BBB. NPs recognized a low-density lipoprotein receptor expressed by SK-N-MC cells and yielded neuroprotective effect on the neuronal degeneration induced by fibrillar Aβ1–42.	[219]
NGF	Liposomes	Cereport trans-ferrin	l-α-PC, DPPC, CL, DSPE-PEG-carboxy.	152-189	from −4.5 to −8.5	31.2 ± 2.8	-	-	-	Covering of NPs with cereport and transferrin was effective in carrying NGF across the BBB and rescued the apoptosis of SK-N-MC cells after neurotoxic treatment with Aβ1–42.	[220]
Fibroblast growth factor	PEG-PLGA NPs	Solanum tubero-sum lectin	PEG-PLGA	118.7	−31,18	69.21	0.0462	-	i.n.	Neuroprotective effect in AD rats was demonstrated.	[221]
Pituitary adenylate cyclaseactivating Polypeptide (PACAP38) coupled to a docosahexaenoic acid (DHA: an ω-3 polyunsaturated fatty acid)	Pep-lipid nanostructures and liquid crystalline nanocarriers	-	PACAP-DHA/MO (nonlamellar lipid monoolein)/DHA/vitamin E/VPGS-PEG1000 and MO/DHA/vitamin E/VPGS-PEG1000	~100	-	-	-	-	-	Multicompartment nanocarriers with PACAP and DHA are promising therapy strategies for neurodegenerative disorders. PACAP is neuropeptide ligand of the PAC1 membrane receptor (a class B GPCR). PACAP demonstrate antiapoptotic effects in neuronal and non-neuronal cells, neuroprotive and neurotransmitter properties, can act as neurotrophic factor, attenuates Aβ amyloid toxicity. DHA plays a crucial role in the control of the phospholipid membrane organization, act as lipid trophic factor, demonstrated antiapoptotic, neuroprotective effect, take part in inflammatory cell signaling. Vitamin E reduces oxidative stress, demonstrates anti-inflammatory, cardio and neuroprotective effects, inhibit lipid peroxidation.	[222]
-	Amphiphilic yellow-emissive carbon dots (Y-CDs)	-	Citric acid, o-phenylenediamine.	3.4 ± 1.0	−15.3	-	-	-	Injected into the heart.	Y-CDs entered cells to inhibit the overexpression of human amyloid precursor protein and β-amyloid.	[223]
-	Nanosystem CB-Gd-Cy5.5	Holera toxin B subunit (CB)	Chelated gadolinium (Gd), Cyanine5.5 (Cy5.5).	110	−11.0	-	-	-	i.n.	The nanosystem was accumulated in the hippocampus and demonstrated good magnetic resonance imaging capability satisfying the monitoring of AD at the different stages.	[224]
-	Fe_3_O_4_ NPs loaded with PEG-PLGA nano composite	Anti-transferrin mono clonal antibody (OX26) receptor	Fe_3_O_4_NPs, PEG, PLGA.	95	-	76.2	18.1	-	-	The significant in vitro drug release and cell viability were shown.	[225]
-	Carboxyl magnetic nano containers	-	Fluorescent Carboxyl Magnetic Particles, Yellow, 1% *w*/*v*	700–900	-	-	-	-	i.v.	The magnetic NPs crossed the normal BBB in mice after subjection to external electromagnetic fields of 28 mT (0.43 T/m) and 79.8 mT (1.39 T/m).	[226]
-	Gold NPs	-	Gold NPs suspension	5	−47.7 ± 10.9	-	-	-	i.p.	The NPs improved the acquisition and retention of spatial learning and memory in Aβ treated rats. Expression of BDNF, cAMP, CREB and stromal interaction molecules, e.g., STIM1 and STIM2 was increased.	[227]

AD—Alzheimer’s disease; Aβ—amyloid-β peptide; BBB—Blood–brain barrier; NC—nanocarrier; NPs—nanoparticles; PS—particle size; ZP—zeta potential; EE—drug entrapment efficiency; LC—loading capacity; PDI—polydispersity index; i.m.—Intramuscular route of administration; i.n.—intranasal route of administration; i.p.—intraperitoneal route of administration; i.v.—intravenous route of administration; p.o.—per oral route of administration; t.d.—transdermal route of administration; BDNF—brain-derived neurotrophic factor; CL—cholesterol; CS—chitosan; DPPC—1,2-dipalmitoyl-sn-glycero-3- phosphocholine; DMPC—1,2-dimyristoyl-sn-glycero-3-phosphocholine; DSPC—1,2-distearyl-sn-glycero-3-phosphocholine; DSPE-PEG—1,2-distearoyl-sn-glycero-3- phosphoethanolamine-N-[carboxy(polyethylene glycol); Gal—galantamine hydrobromide; Hup—huperzine A; Lf—lactoferrin; NAP—neuroprotective peptide NAPVSIPQ; NGF—neuron growth factor; PC—phosphatidylcholine; PEG—(poly(ethylene glycol); PEG-PCL—poly(ethylene glycol)-co-poly(ε-caprolactone) copolymer; PEPEG—1,2-dipalmitoyl-sn-glycero-3-phosphoethanol- amine-N-[methoxy(polyethylene glycol)-2000]; PLA—poly(lactic acid); PLGA—poly(lactic-co-glycolic acid); PVA—polyvinyl alcohol; siRNA—small interfering RNAs; Plu—pluronic; Riv—rivastigmine;.

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
