# Peer review of "Nano Carrier Drug Delivery Systems for the Treatment of Neuropsychiatric Disorders: Advantages and Limitations"

_molecules, 2020, doi:10.3390/molecules25225294_

Round 1
Reviewer 1 Report
Dear Authors,
Thank you for your submission to MDPI Molecules. Your review article is well researched, well written, and poses a helpful perspective to the field of nanomedicine for drug delivery to the brain. Your article is timely, covers a broad range of issues in the field of drug delivery for neuropsychiatric disorders.
A few notes / comments:
Please note that none of these items are absolutely required for publication, but their inclusion would significantly improve the paper.
-- It seems that Table 1 would be improved with the addition of references supporting the statements made. Not absolutely critical, but possibly helpful.
--Can the authors define "GBP" in line 147?
--While there is significant discussion of various drugs, particles, and drug particle combinations for treatment, a few topics seem to be missing. If possible, addition of any of the following sections may be interesting to readers and could improve the overall scope and quality of the review:
Particle Transport in the Brain (passive or active / magnetic / ultrasound guided);
Drug-Particle Conjugate Interactions with Various Types of Neural Cells / Tissues;
Uptake by Neurons and Other Neural Cells Types;
Methods for Targeting Specific Cells, Cell Types, or Brain Regions;
Testing Methods for Neurotoxicity.
--It may be informative if the authors included a table listing pros and cons of nanocarrier delivery for nanocarrier drug delivery systems as compared with other methods of delivery drugs for neuropsychiatric disorders.
--Some figures showing the various drug delivery methods and pathways could be informative and provide the reader with a clearer understanding of the various concepts and findings being described. Particularly helpful would be images that depict the various modes of delivery via oral, injection, brain injection / delivery, and nasal delivery.
--Some figures showing how various nanocarriers interact with neural tissues / cells could also be helpful.
--It would also be helpful if the article contained a table of contents at the beginning so that readers could skip to relevant sections if they liked.
Author Response
Reviewer
Dear Reviewer! Thank you for your detailed analysis of our manuscript. We tried to accept your comments and to make necessary adjustments. We hope that recommended corrections will significantly improve our review and make it more interesting for readers.
All corrections are marked by yellow.
-- It seems that Table 1 would be improved with the addition of references supporting the statements made. Not absolutely critical, but possibly helpful.
Thank you! We added required references into the Table. Now it named Table 2, because we added one more table. Page 10.
--Can the authors define "GBP" in line 147?
It was our misprint. We meant BBB – blood brain barrier. This abbreviation was corrected. Page 19, line 257
--While there is significant discussion of various drugs, particles, and drug particle combinations for treatment, a few topics seem to be missing. If possible, addition of any of the following sections may be interesting to readers and could improve the overall scope and quality of the review:
Particle Transport in the Brain (passive or active / magnetic / ultrasound guided);
Drug-Particle Conjugate Interactions with Various Types of Neural Cells / Tissues;
Uptake by Neurons and Other Neural Cells Types;
Methods for Targeting Specific Cells, Cell Types, or Brain Regions;
Testing Methods for Neurotoxicity.
Thank you for your useful remark! We added necessary information into the part “Drug transport through blood brain barrier”. We described common ways of nanoparticles uptake by cells and characterized strategies of neurons and glial cells targeting. We hope that included information will increase the quality of the manuscript. Page 8, line 185-224
--It may be informative if the authors included a table listing pros and cons of nanocarrier delivery for nanocarrier drug delivery systems as compared with other methods of delivery drugs for neuropsychiatric disorders.
We added required table (Table 1) representing comparative analysis of nanoparticles with other methods of drug delivery (intranasal administration, transdermal administration, oral administration, intravenous administration) into the introduction. Page 4.
--Some figures showing the various drug delivery methods and pathways could be informative and provide the reader with a clearer understanding of the various concepts and findings being described. Particularly helpful would be images that depict the various modes of delivery via oral, injection, brain injection / delivery, and nasal delivery.
--Some figures showing how various nanocarriers interact with neural tissues / cells could also be helpful.
Thank you for your commentary! We added the recommended illustration into our manuscript. Figure 1 describes the main pathways of nanoparticles entering the brain: mechanisms for BBB penetration and transport across the cell membrane. We added this figure after the part “Drug transport through blood brain barrier”. Page 7.
--It would also be helpful if the article contained a table of contents at the beginning so that readers could skip to relevant sections if they liked.
The table of contents was added before introduction. Page 2-3, line 61-91.
Reviewer 2 Report
The manuscript presents a detailed review on nanoparticles for potential treatment of neuropsychiatric disorders. The following revisions can be done towards broader readership and impact of the article.
1) Presently the Abstract does not specify which questions will be elaborated in the paper. The different chapters of the review should be summarized in the abstract so that the reader knows what the content of the article is. Is there a particular unsolved question on which you will focus?
2) Introduction, lines 55-57:
References should be included about liquid crystalline nanoparticles with neuroprotective properties, e.g.
- ACS Omega, 2019, 4, 3061–3073, DOI: 10.1021/acsomega.8b03101 ;
3) Figures can be created to illustrate which signaling molecules and receptor pathways represent targets in the therapies of anxiety disorders, schizophrenia, AD, etc.
Alternatively, it is recommended to reproduce with permission two or three illustrations from published articles. The presented information is summarized in Tables. However, figures are not included.
4) Page 22, lines 360-370:
Published works on in vitro carriers of neurotrophic protein and peptides encapsulated alone or in combinations with other neuroprotective molecules should be cited, e.g.
- ChemNanoMat 2019, 5, 1381–1389, DOI: 10.1002/cnma.201900468 ;
Author Response
Reviewer 2
Dear Reviewer 2! Thank you for your relevant commentaries and recommendations. We tried to improve our manuscript according to your comments. We hope that integrated corrections will help to make our review more accessible and interesting for specialists in this field.
All corrections are marked by yellow.
The manuscript presents a detailed review on nanoparticles for potential treatment of neuropsychiatric disorders. The following revisions can be done towards broader readership and impact of the article.
1) Presently the Abstract does not specify which questions will be elaborated in the paper. The different chapters of the review should be summarized in the abstract so that the reader knows what the content of the article is. Is there a particular unsolved question on which you will focus?
We rewrite the abstract (page 1, line 18-33):
Neuropsychiatric diseases are one of the main causes of disability, affecting millions of people. Various drugs are used for its treatment, although no effective therapy has been found yet. The blood brain barrier (BBB) significantly complicates drugs delivery to the target cells in the brain tissues. One of the problem-solving methods is the usage of nanocontainer systems. In this review we summarized the data about nanoparticles drug delivery systems and their application for the treatment of neuropsychiatric disorders. Firstly we described and characterized types of nanocarriers: inorganic nanoparticles, polymeric and lipid nanocarriers, their advantages and disadvantages. We discussed ways to interact with nerve tissue and methods of BBB penetration. We provided a summary of nanotechnology-based pharmacotherapy of schizophrenia, bipolar disorder, depression, anxiety disorder, Alzheimer’s disease, where development of nanocontainer drugs derives the most active. We described various experimental drugs for the treatment of Alzheimer's disease that include vector nanocontainers targeted on β-amyloid or tau-protein. Integrally, nanoparticles can substantially improve the drug delivery as its implication can increase BBB permeability, the pharmacodynamics and bioavailability of applied drugs. Thus, nanotechnology is anticipated to overcome the limitations of existing pharmacotherapy of psychiatric disorders and to effectively combine various treatment modalities in that direction.
2) Introduction, lines 55-57:
References should be included about liquid crystalline nanoparticles with neuroprotective properties, e.g.
- ACS Omega, 2019, 4, 3061–3073, DOI: 10.1021/acsomega.8b03101 ;
Thank you! We added this reference in the introduction (page 5, line 119, reference number 3).
3) Figures can be created to illustrate which signaling molecules and receptor pathways represent targets in the therapies of anxiety disorders, schizophrenia, AD, etc.
Alternatively, it is recommended to reproduce with permission two or three illustrations from published articles. The presented information is summarized in Tables. However, figures are not included.
Thank you for your relevant recommendation! We decided to create the picture representing the main mechanisms of Alzheimer’s disease (amyloid aggregation, tau fibrillation, impairments of acetylcholine, glutamate systems, and mitochondria functioning) and therapy methods, targeted to these mechanisms (page 45), as it is described in the manuscript. We haven’t illustrated pathogenic pathways for other disorders presented in the review, because that information wasn’t expound in our article. If necessary, we could add the data about pathogenesis of mental disorders in the review and illustrate it with detailed figures.
4) Page 22, lines 360-370:
Published works on in vitro carriers of neurotrophic protein and peptides encapsulated alone or in combinations with other neuroprotective molecules should be cited, e.g.
- ChemNanoMat 2019, 5, 1381–1389, DOI: 10.1002/cnma.201900468 ;
We added required reference in the Table 4 “Summary of nanotechnology-based systems applied in the treatment of Alzheimer’s disease” (page 63-64, reference number 222).